# The molecular appearance of native TRPM7 channel complexes identified by high-resolution proteomics

Astrid Kollewe[1†], Vladimir Chubanov[2†], Fong Tsuen Tseung[2], Leonor Correia[2], Eva Schmidt[2], Anna Rössig[2], Susanna Zierler[2,3], Alexander Haupt[1], Catrin Swantje Müller[1], Wolfgang Bildl[1], Uwe Schulte[1,4], Annette Nicke[2], Bernd Fakler[1,4]*, Thomas Gudermann[2,5]*

[1]Institute of Physiology II, Faculty of Medicine, University of Freiburg, Freiburg, Germany; [2]Walther-Straub Institute of Pharmacology and Toxicology, LMU Munich, Munich, Germany; [3]Institute of Pharmacology, Johannes Kepler University Linz, Linz, Austria; [4]Signalling Research Centres BIOSS and CIBSS, Freiburg, Germany; [5]German Center for Lung Research, Munich, Germany

*For correspondence:
bernd.fakler@physiologie.uni-freiburg.de (BF);
Thomas.Gudermann@lrz.uni-muenchen.de (TG)

[†]These authors contributed equally to this work

**Competing interest:** The authors declare that no competing interests exist.

**Abstract** The transient receptor potential melastatin-subfamily member 7 (TRPM7) is a ubiquitously expressed membrane protein consisting of ion channel and protein kinase domains. TRPM7 plays a fundamental role in the cellular uptake of divalent cations such as $Zn^{2+}$, $Mg^{2+}$, and $Ca^{2+}$, and thus shapes cellular excitability, plasticity, and metabolic activity. The molecular appearance and operation of TRPM7 channels in native tissues have remained unresolved. Here, we investigated the subunit composition of endogenous TRPM7 channels in rodent brain by multi-epitope affinity purification and high-resolution quantitative mass spectrometry (MS) analysis. We found that native TRPM7 channels are high-molecular-weight multi-protein complexes that contain the putative metal transporter proteins CNNM1-4 and a small G-protein ADP-ribosylation factor-like protein 15 (ARL15). Heterologous reconstitution experiments confirmed the formation of TRPM7/CNNM/ARL15 ternary complexes and indicated that complex formation effectively and specifically impacts TRPM7 activity. These results open up new avenues towards a mechanistic understanding of the cellular regulation and function of TRPM7 channels.

## Editor's evaluation

This work will be interesting to people studying TRP family ion channels and more generally, cellular ion homeostasis. It is the first to identify interacting protein partners of the cation channel TRPM7, a key regulator of cellular $Mg^{2+}$ and $Zn^{2+}$ homeostasis, and reveals functional coupling between TRPM7, a putative magnesium transporter, and a small G protein.

## Introduction

Transient receptor potential melastatin-subfamily member 7 (TRPM7) encodes a bi-functional protein with a transient receptor potential (TRP) ion channel domain fused to a C-terminal α-type serine/threonine-protein kinase (reviewed in *Chubanov et al., 2018*; *Fleig and Chubanov, 2014*; *Ryazanov et al., 1997*). Among all other known channels and kinases, only its homologue TPRM6 shows a similar design (*Ryazanov et al., 1997*; *Chubanov and Gudermann, 2014*).

TRPM7 is involved in various cellular processes such as homeostatic balance, cell motility, proliferation, differentiation, and regulation of immune responses (*Chubanov et al., 2018*; *Fleig and*

*Chubanov, 2014*; *Ryazanov et al., 1997*). Genetic deletion of *Trpm7* in mice is embryonically lethal, and tissue-specific null mutants have shown defects in cardiac and renal morphogenesis, organismal $Zn^{2+}$, $Mg^{2+}$, and $Ca^{2+}$ homeostasis, thrombopoiesis, and mast cell degranulation (*Mittermeier et al., 2019*; *Chubanov et al., 2004*; *Jin et al., 2008*; *Sah et al., 2013b*; *Sah et al., 2013a*; *Jin et al., 2012*; *Stritt et al., 2016*; *Abiria et al., 2017*; *Schmitz et al., 2003*). Besides, TRPM7 has emerged as a promising therapeutic target for numerous pathophysiological conditions (*Chubanov et al., 2018*; *Fleig and Chubanov, 2014*; *Ryazanov et al., 1997*; *Hofmann et al., 2014*; *Aarts et al., 2003*; *Hermosura et al., 2005*).

The channel-coding segment of TRPM7 comprises six transmembrane helices with a pore-loop sequence between S5 and S6 (*Figure 1A*, *Duan et al., 2018*; *Mederos y Schnitzler et al., 2008*). Four subunits assemble to form constitutively active channels highly selective for divalent cations such as $Zn^{2+}$, $Ca^{2+}$, and $Mg^{2+}$ (*Nadler et al., 2001*; *Runnels et al., 2001*; *Monteilh-Zoller et al., 2003*). Free $Mg^{2+}$, the Mg·ATP complex, and phosphatidylinositol-4,5-bisphosphate (PIP$_2$) were described as physiological regulators of the channel activity of TRPM7 (*Nadler et al., 2001*; *Runnels et al., 2002*). While $Mg^{2+}$ or Mg·ATP act as negative regulators, PIP$_2$ appears to be a crucial co-factor of the active channel (*Nadler et al., 2001*; *Runnels et al., 2002*). Mechanistically, however, the effects of $Mg^{2+}$, Mg·ATP, or PIP$_2$ on TRPM7 activity are poorly understood, and most likely, there are additional regulators of TRPM7 function with hitherto unknown molecular identity.

The C-terminal α-kinase domain of TRPM7 acts in two ways: First, it autophosphorylates cytoplasmic residues of TRPM7, and second, it may target a variety of proteins with diverse cellular functions such as annexin A1, myosin II, eEF2-k, PLCγ2, STIM2, SMAD2, and RhoA (*Runnels et al., 2001*; *Dorovkov and Ryazanov, 2004*; *Perraud et al., 2011*; *Clark et al., 2008*; *Romagnani et al., 2017*; *Voringer et al., 2020*; *Faouzi et al., 2017*). In immune cells, the TRPM7 kinase domain has been reported to be clipped from the channel domain by caspases in response to Fas-receptor stimulation (*Desai et al., 2012*). In line with this observation, cleaved TRPM7 kinase was detected in several cell lines and shown to translocate to the nucleus, where it promotes histone phosphorylation (*Krapivinsky et al., 2014*).

The majority of the current knowledge about TRPM7 was derived from in vitro experiments with cultured cells, whereas insights into the operation of both channel and α-kinase activity of TRPM7 in native tissues are limited. We, therefore, investigated the molecular architecture of TRPM7 in rodent brain by using blue native polyacrylamide gel electrophoresis (BN-PAGE) and multi-epitope affinity purifications (ME-APs) in combination with high-resolution quantitative mass spectrometry (MS). These approaches showed that native TRPM7 channels are macromolecular complexes with an apparent size of ≧1.2 MDa and identified proteins CNNM1-4 and ADP-ribosylation factor-like protein 15 (ARL15) as complex constituents. Subsequent functional studies in *Xenopus laevis* oocytes and HEK293 cells suggested ARL15 and CNNM3 as hitherto unrecognised regulators of the TRPM7 ion channel and kinase activity, respectively.

## Results

### ME-AP proteomic analyses of native TRPM7 channels

TRPM7 channels assemble from four subunits (*Fleig and Chubanov, 2014*), each of which is about 1860 aa in length and comprises several distinct domains in its extended intracellular N- and C-termini in addition to a transmembrane channel domain (*Figure 1A*). Unexpectedly, analysis by native gel electrophoresis (BN-PAGE) of TRPM7 channels either endogenous to HEK293 cells or exogenously expressed in these cells via transient transfection, elicited a molecular mass of at least 1.2 MDa considerably exceeding the molecular mass of ~850 kDa calculated for TRPM7 tetramers (*Figure 1B*, upper panel). To see whether this large molecular size is a peculiarity of HEK293 cells, we recapitulated the analysis for TRPM7 channels expressed in mouse brain using a recently developed technique that combines BN-PAGE with cryo-slicing and quantitative mass spectrometry (csBN-MS, *Müller et al., 2019*). In this approach, membrane fractions prepared from the entire mouse brain and solubilised with the mild detergent buffer CL-47 (*Schwenk et al., 2016*; *Schwenk et al., 2012*; *Müller et al., 2010*) are first separated on a native gel, which is subsequently embedded and cut into 300 μm gel slices using a cryo-microtome. In a second step, the protein content of each slice is analysed individually by nanoflow liquid chromatography tandem mass spectrometry (nanoLC-MS/MS), providing

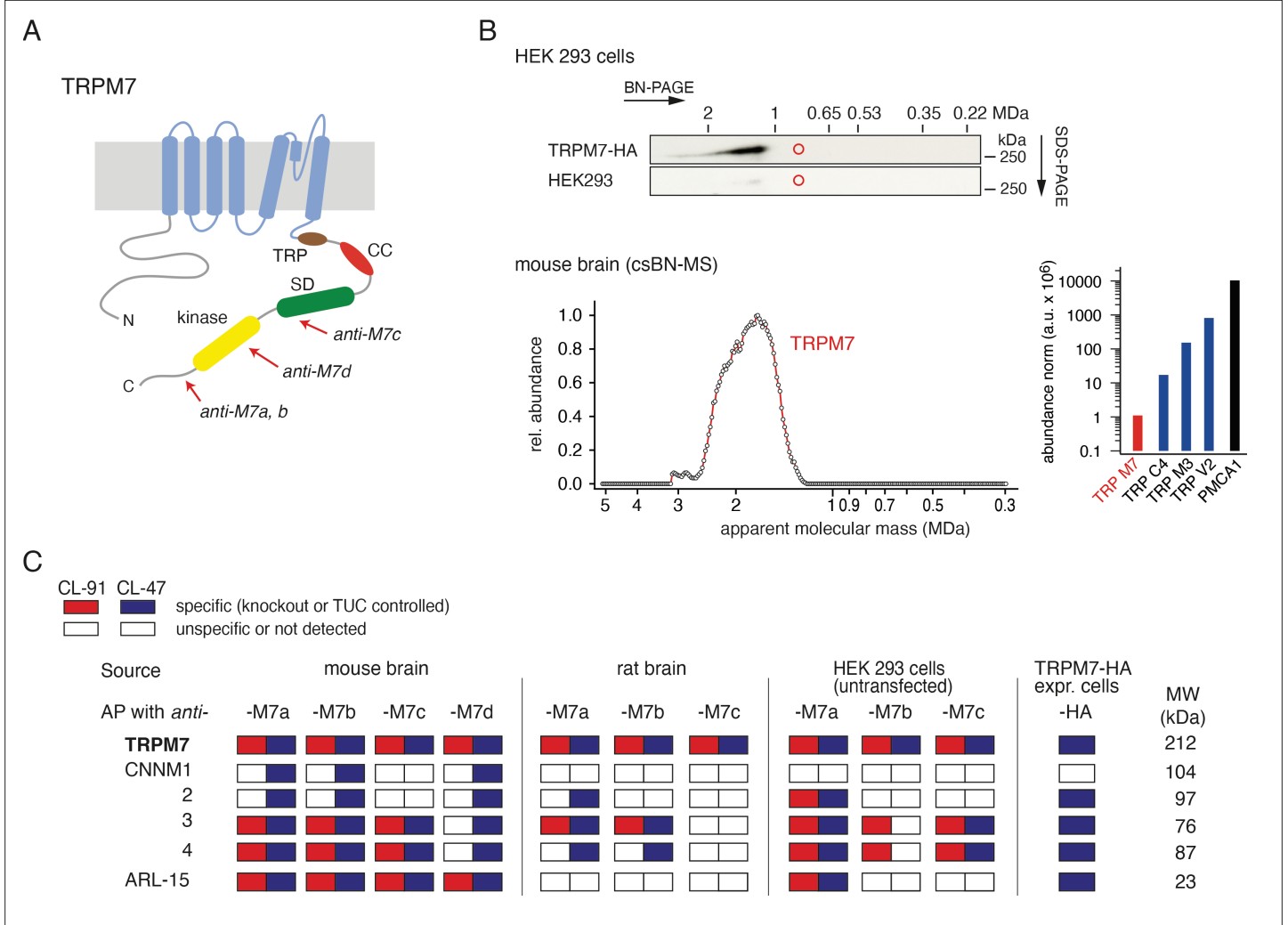

**Figure 1.** Protein constituents of native transient receptor potential melastatin-subfamily member 7 (TRPM7) channels identified by multi-epitope antibody-based affinity purification (ME-AP) proteomics. (**A**) Topology and localisation of the *anti*-TRPM7 antibodies used for ME-APs. Established hallmark domains of TRPM7 are colour-coded, TRP (transient receptor potential domain, brown), CC (coiled-coil domain, red), kinase (kinase domain, yellow), SD (serine/threonine-rich substrate domain of kinase(s), green). (**B**) *Upper panel*: Two-dimensional gel separation of TRPM7 channels in CL-47 solubilised membrane fractions of HEK293 cells with (*upper panel*) or without (*lower panel*) transfection of HA-tagged *Trpm7*, Western-probed with an *anti*-TRPM7 antibody (Materials and methods). Size (blue native polyacrylamide gel electrophoresis [BN-PAGE]) and molecular weight (SDS-PAGE) are as indicated. *Lower panel*: Abundance-mass profile of TRPM7 obtained by cryo-slicing blue native mass spectrometry (csBN-MS) in a CL-47 solubilised membrane fraction from adult mouse brain (a total of 192 gel slices). Inset: Abundance of the indicated proteins in the mouse brain. Note the large apparent molecular mass of the native TRPM7 channel in both culture cells and mouse brain, markedly exceeding the mass calculated for tetrameric channel assemblies (about 850 kDa, red circles). (**C**) Table summarising the results of all *anti*-TRPM7 APs performed with the indicated antibodies on membrane fractions prepared from rodent brain and cultured HEK293 cells. Solubilisation conditions and specificity of purification of the listed proteins determined by comparison with stringent negative controls are colour-coded as given in the upper left; MW is indicated on the right. TUC refers to series of APs with target-unrelated control antibodies. Note that TRPM7 channels co-assemble with all CNNM family members and ADP-ribosylation factor-like protein 15 (ARL15) in the brain and HEK293 cells.

The online version of this article includes the following figure supplement(s) for figure 1:

**Figure supplement 1.** The specificity of an *anti*-transient receptor potential melastatin-subfamily member 7 (TRPM7) mouse monoclonal antibody in Western blot assessment of the recombinant TRPM6 and TRPM7 proteins.

information on both the identity and amount of the proteins in each slice; noteworthy, protein amounts are determined with a dynamic range of up to four orders of magnitude (*Müller et al., 2010*; *Schwenk et al., 2010*; *Bildl et al., 2012*). As illustrated in *Figure 1B*, lower panel, csBN-MS analysis of mouse brain membranes detected the TRPM7 protein with an apparent molecular mass between 1.2 and 2.6 MDa, comparable to the results obtained from HEK293 cells (*Figure 1B*, upper panel). Moreover, the

determination of the total protein amount by signal integration over all slices showed that TRPM7 levels in the brain are rather low compared to other members of the TRP family of proteins. Thus, the abundance of TRPM7 is about one to three orders of magnitude below that obtained for TRPC4, TRPM3, or TRPV2 (*Figure 1B*, lower right).

Together, these results indicated that native TRPM7 complexes exceed the predicted molecular size of bare tetrameric assemblies in different cellular environments suggesting that the rather simplistic view on the molecular make-up of native TRPM7 channel complexes has to be revised.

To identify proteins that may co-assemble with TRPM7, we used affinity purifications with multiple antibodies targeting distinct epitopes of the TRPM7 protein (*Figure 1A*, *Figure 1—figure supplement 1*) and evaluated the respective eluates of HEK293 cells and rodent brains by high-resolution quantitative MS analysis (ME-APs, *Schwenk et al., 2016*; *Schwenk et al., 2012*; *Müller et al., 2010*; *Schwenk et al., 2010*). HEK293 cells were selected because these cells are widely used for the functional assessment of endogenous and overexpressed TRPM7. The brain was chosen since TRPM7 plays a critical role in neurological injuries and synaptic and cognitive functions (*Aarts et al., 2003*; *Sun et al., 2009*; *Liu et al., 2018*). For these ME-APs, membrane fractions prepared either from whole brains of adult mice and rats or from WT HEK293 cells were solubilised with detergent buffers of mild (CL-47) or intermediate (CL-91) stringency (*Schwenk et al., 2012*; *Müller et al., 2010*; *Schwenk et al., 2010*) prior to TRPM7 purification. TRPM7 was also affinity-isolated from HEK293 cells transiently (over)-expressing C-terminally HA-tagged TRPM7 using an *anti*-HA antibody.

In all APs, TRPM7 could be reliably detected under both solubilisation conditions (*Figure 1C*) with MS-identified peptides covering a large percentage of the primary sequence of TRPM7 in samples from mouse brain as well as from HEK293 cells (77% and 98%, respectively).

All other proteins identified in the ME-APs were evaluated for specificity and consistency of their co-purification with TRPM7 based on protein amounts determined by label-free quantification (see Materials and methods section). The specificity of co-purification was assessed by comparing protein amounts in APs targeting TRPM7 with protein amounts obtained with stringent negative controls. Thus, (i) APs with five different target-unrelated control (TUC) antibodies were used as negative controls for *anti*-TRPM7 APs from rodent brain, (ii) *anti*-TRPM7 APs from a *TRPM7⁻/⁻* HEK293 cell line (*Abiria et al., 2017*) served as negative controls for *anti*-TRPM7 APs from WT HEK293 cells, and (iii) HEK293 cells heterologously expressing TRPM7-myc were used as negative

**Table 1.** Protein constituents of native transient receptor potential melastatin-subfamily member 7 (TRPM7) channels identified by multi-epitope affinity purifications (ME-APs).

| Protein ID | Acc. No. UniProtKB | Name | Primary function | Rel. abundance CL-47 | CL-91 |
|---|---|---|---|---|---|
| TRPM7 | Q923J1 | TRP channel M7 | Ion channel | = | = |
| CNNM1 | Q0GA42 | Transporter CNNM1, Cyclin-M1 | Potential transporter | << | |
| CNNM2 | Q5U2P1 | Transporter CNNM1, Cyclin-M2 | Potential transporter | < | << |
| CNNM3 | Q32NY4 | Transporter CNNM1, Cyclin-M3 | Potential transporter | < | << |
| CNNM4 | Q69ZF7 | Transporter CNNM1, Cyclin-M4 | Potential transporter | < | << |
| ARL15 | Q8BGR6 | ADP-ribosylation factor-like protein 15 | Unknown | = | << |
| TP4A1[†] | Q93096 | Protein tyrosine phosphatase type IVA 1 | Enzyme | <<< | << |
| TP4A3[##] | Q9D658 | Protein tyrosine phosphatase type IVA 3 | Enzyme | | << |
| TRPM6[###] | Q9BX84 | TRP channel M6 | Ion channel | <<< | << |

Notes: Relative abundance refers to the amount of TRPM7 as a reference and was classified as follows: = when between 0.33-fold and 3.3-fold of reference, < when between 0.033-fold and 0.33-fold of reference, << when between 0.0033-fold and 0.033-fold of reference, and <<< when less than 0.0033-fold of the reference amount.

■ Transmembrane proteins; ■ cytoplasmic proteins.

[†]Co-purified from HEK293 cells with *anti*-M7a (CL-47) and with *anti*-M7c (CL-91); [##]co-purified with anti-M7c from rat brain membranes (CL-91); [###]co-purified with anti-M7a from HEK293 cells (CL-47, CL-91).

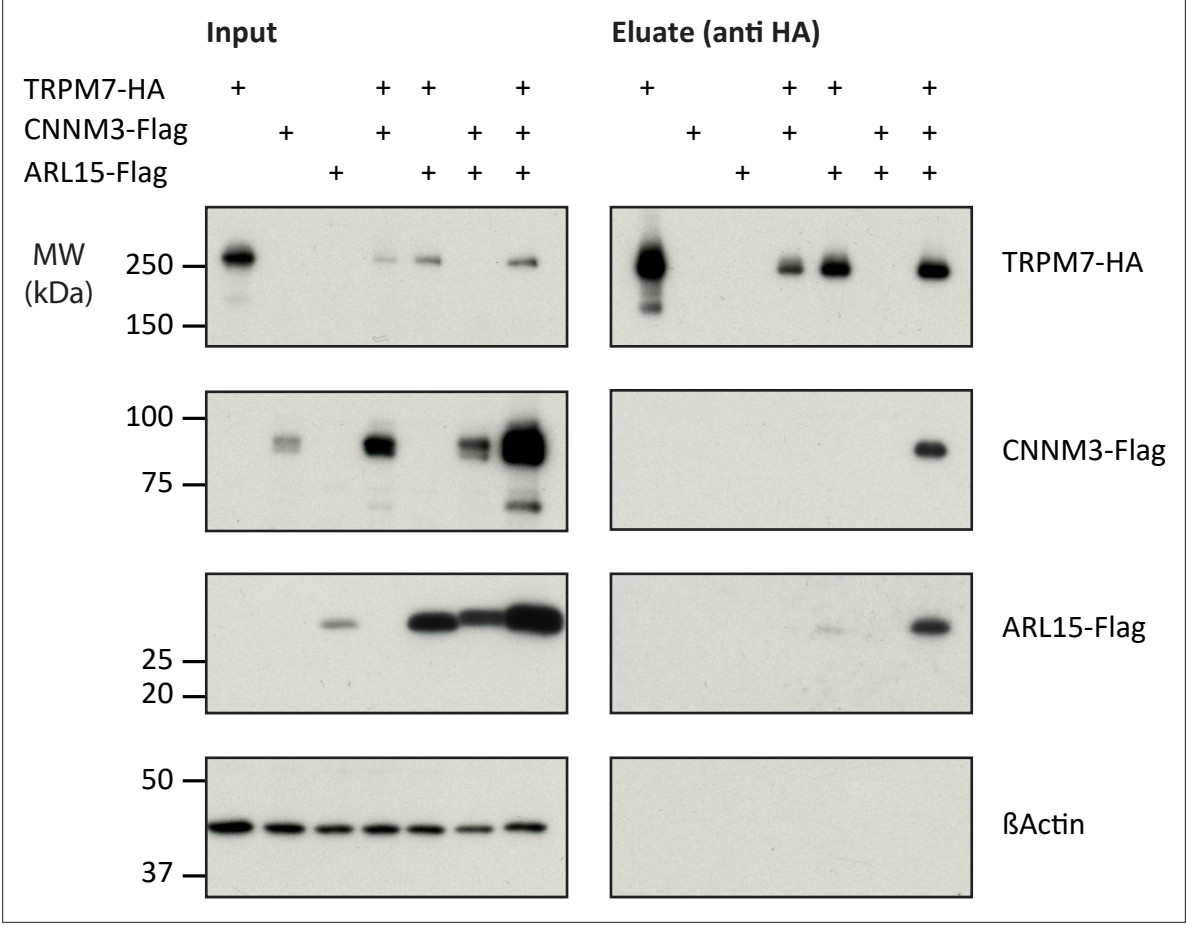

**Figure 2.** Heterologous reconstitution of transient receptor potential melastatin-subfamily member 7 (TRPM7) complexes in HEK293 cells. Affinity purifications (APs) with *anti*-HA antibody from CL-47 solubilised membrane fractions of *TRPM7⁻ᐟ⁻* HEK293 cells transiently expressing the proteins indicated above. Input and eluates of the distinct APs were separated by SDS-PAGE and Western-probed with *anti*-Flag, *anti*-HA, and *anti*-β-actin antibodies. Molecular weight (MW) is marked on the left.

The online version of this article includes the following figure supplement(s) for figure 2:

**Figure supplement 1.** Heterologous reconstitution of transient receptor potential melastatin-subfamily member 7 (TRPM7) complexes in HEK293 cells.

controls for *anti*-HA APs from HEK293 cells overexpressing TRPM7-HA. A protein was considered consistently co-purified if detected in APs with at least two antibodies under the same solubilisation condition. Together, these specificity and consistency criteria identified five proteins as high-confidence interaction partners of TRPM7: ARL15 and the cyclin M family proteins CNNM1-4, putative Mg²⁺ transporters (*Figure 1C*, *Table 1*). Neither of these proteins was detected in any of the negative controls. Moreover, they were not only consistently co-purified with several antibodies but with the exception of CNNM1 also from both rodent brain and HEK293 cells. Comparison of the degree of association under the two solubilisation conditions revealed that the interaction between TRPM7, ARL15, and CNNMs was weakened by the more stringent detergent CL-91 (*Figure 1C*, *Table 1*).

Next, we verified the identified interactions between TRPM7, ARL15, and CNNM1-4 in co-expression experiments performed in *TRPM7⁻ᐟ⁻* HEK293 cells (*Figure 2*). Flag-tagged ARL15 and CNNM proteins could be specifically and robustly co-purified with HA-tagged TRPM7 in *anti*-HA APs when all three proteins were present, whereas the association was markedly less efficient when ARL15-Flag or CNNM-Flag were co-expressed with TRPM7-HA alone (*Figure 2*, *Figure 2—figure supplement 1*). These results corroborated the ME-AP results from the rodent brain and strongly suggested the formation of ternary complexes containing TRPM7, ARL15, and CNNM proteins.

## Effects of CNNM3 and ARL15 on TRPM7 channel activity

To investigate if the assembly of TRPM7 with ARL15 and CNNM proteins modified TRPM7 function, we studied their effect(s) on TRPM7 currents by co-expression in *X. laevis* oocytes. This approach allows co-expression of defined protein ratios by cRNA injection and, therefore, is widely used for functional assessment of ion channel complexes, including functional interaction of TRPM7 with TRPM6 (*Chubanov et al., 2018*; *Chubanov et al., 2004*). The two-electrode voltage clamp (TEVC) measurement in *Figure 3A* illustrates a typical current-voltage (I-V) relationship of constitutively active TRPM7 channels characterised by steep outward rectification and very small inward currents over the whole range of negative membrane potentials (*Nadler et al., 2001*). Co-expression of TRPM7 and CNNM3, the most efficiently co-purified CNNM protein (*Figure 1C*), neither changed the shape of the I-V relationship nor current amplitudes. In contrast, ARL15 effectively suppressed constitutive TRPM7 currents in a concentration-dependent manner, as deduced from experiments with increasing amounts of ARL15 (*Figure 3B and C*). Oocytes co-expressing all three proteins TRPM7, CNNM3, and ARL15 did not exhibit TRPM7 currents, similar to the co-expression of TRPM7 and ARL15 (*Figure 3A*). The suppressive effect was specific for TRPM7, as co-expressed ARL15 did not inhibit another TRP channel, TRPV1, in an analogous experiment (*Figure 3—figure supplement 1*). Consistently, co-expression of TRPM7 with another ARL family member, ARL8A (*Gillingham and Munro, 2007*), did not affect TRPM7 currents (*Figure 3—figure supplement 2*).

Next, we examined if the interference of ARL15 with the TRPM7 function was due to reduced expression levels or altered membrane localisation. Western blot analysis of oocytes injected with *Trpm7* or *Trpm7* and *Arl15* cRNAs did not reveal any change in the expression level of TRPM7 protein (*Figure 3D*). Using immunofluorescence staining with the *anti*-M7d antibody, we detected TRPM7 at the cell surface of oocytes injected with *Trpm7* but not in uninjected oocytes (*Figure 3E*). Notably, the TRPM7 signal was similarly detectable at the cell surface of oocytes co-expressing TRPM7 and ARL15 (*Figure 3E*).

TRPM7 inward currents at negative membrane potentials are small, and, consequently, quantification of the comparably large outward currents is commonly used for functional assessment of the TRPM7 channel activity. Nevertheless, we asked whether TRPM7 inward currents could be equally suppressed by ARL15 (*Figure 3—figure supplement 3A, B*). This analysis revealed that ARL15 acted similarly on inward and outward TRPM7 currents, suggesting that ARL15 elicited a general block of the TRPM7 channel.

To obtain further insight into the functional interaction of ARL15 with TRPM7, we investigated whether the kinase activity of TRPM7 is necessary for the inhibitory effect of ARL15. To this end, we examined oocytes expressing a kinase-dead TRPM7 mutant (K1646R, *Nadler et al., 2001*; *Runnels et al., 2002*) and observed that the K1646R mutation did not change the sensitivity of TRPM7 for the inhibitory effect of ARL15 (*Figure 3—figure supplement 3C*).

Finally, we investigated whether ARL15 could also regulate TRPM7 channels in mammalian cells. Using the patch-clamp technique, we measured endogenous TRPM7 currents in HEK293 cells. Similar to previous reports (*Chubanov et al., 2004*; *Ferioli et al., 2017*), removing intracellular $Mg^{2+}$ by using a pipette solution free of divalent cations induced endogenous TRPM7 currents (*Figure 3—figure supplement 4*). Transient expression of ARL15 however caused a significant reduction of these TRPM7 currents (*Figure 3—figure supplement 4*).

Collectively, these results suggest that the inhibitory effect of ARL15 on TRPM7 currents is specific and concentration-dependent.

## Impact of CNNM3 on TRPM7 $Mg^{2+}$ currents and kinase activity

Given the crucial role of TRPM7 and CNNM proteins in membrane $Mg^{2+}$ transport (*Mittermeier et al., 2019*; *Schmitz et al., 2003*; *Funato and Miki, 2019*), we asked whether CNNM3 would specifically affect TRPM7 $Mg^{2+}$ currents rather than exerting a general (i.e., ARL15-like) effect. To this end, we conducted TEVC measurements with TRPM7-expressing oocytes using external saline containing 3 mM $Mg^{2+}$ (instead of 3 mM $Ba^{2+}$ in *Figure 3A*), implying that at negative membrane potentials, the TRPM7 channel should primarily exhibit $Mg^{2+}$ currents under such experimental conditions (*Nadler et al., 2001*). TRPM7 expressing oocytes displayed characteristic TRPM7 currents with a very small inward $Mg^{2+}$ component, which was suppressed by co-expression of ARL15 (*Figure 4A and B*) in accord with previous experiments (*Figure 3—figure supplement 3A, B*). In contrast, co-expression of CNNM3 did not change the properties of the TRPM7 channel (*Figure 4C and D*).

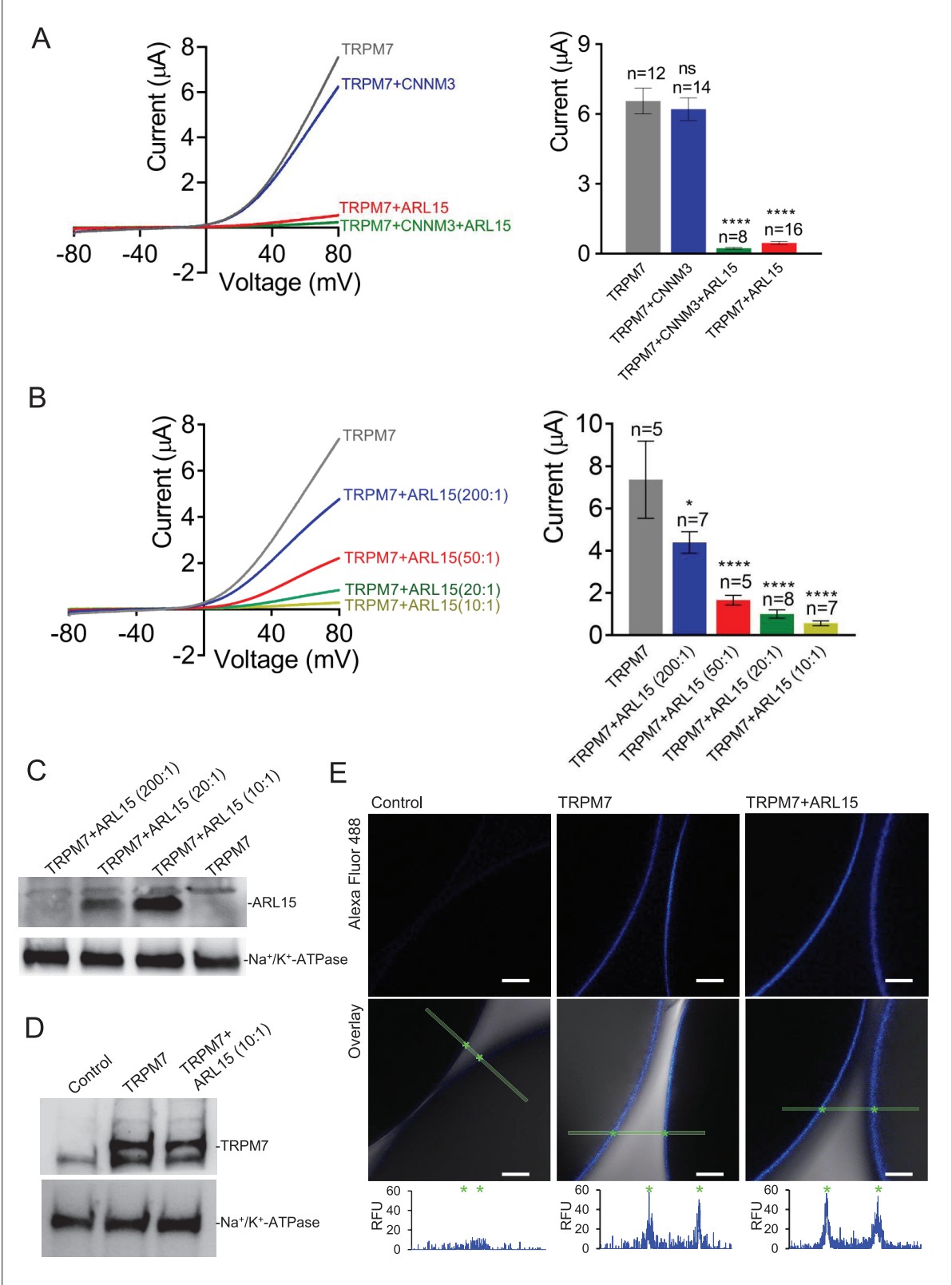

**Figure 3.** Heterologous expression of transient receptor potential melastatin-subfamily member 7 (TRPM7) in *Xenopus* oocytes. (**A, B**) Two-electrode voltage clamp (TEVC) measurements of TRPM7 currents. (**A**) *Left panel*: Representative current-voltage (I-V) relationships of TRPM7 currents measured in oocytes expressing TRPM7 alone or TRPM7 with CNNM3 or ADP-ribosylation factor-like protein 15 (ARL15) (cRNAs ratio 2:1), and TRPM7 with CNNM3 and ARL15 (cRNAs ratio 2:1:1). *Right panel*: Current amplitudes (mean ± standard error of the mean [SEM]) at +80 mV in measurements shown on the

*Figure 3 continued on next page*

*Figure 3 continued*

left. Two independent batches of injected oocytes (n = 8–16) were examined. *p < 0.05; ****p < 0.0001 (ANOVA). (**B**) *Left panel*: Representative I-V relationships of TRPM7 currents measured in oocytes expressing TRPM7 or co-expressing TRPM7 with ARL15 at the indicated ratios of injected cRNAs. *Right panel*: Current amplitudes (mean ± SEM) at +80 mV in measurements shown on the left. Two independent batches of injected oocytes (n = 5–7) were examined. *p < 0.05; ****p < 0.0001 (ANOVA). (**C**) Western blot analysis of ARL15 expression using the *anti*-Myc antibody in total lysates of oocytes injected with *Trpm7* or *Trpm7* and *Arl15* cRNAs (ratios 200:1, 20:1, and 10:1). Representative results are shown for two independent experiments. *Anti*-$Na^+/K^+$-ATPase antibody was used for loading controls. (**D**) Western blot analysis of TRPM7 expression using the *anti*-M7d antibody in total lysates of oocytes injected with *Trpm7* or *Trpm7* and *Arl15* cRNAs (ratio 10:1). *Anti*-$Na^+/K^+$ ATPase antibody was used for loading controls. Representative results are shown for two independent experiments. (**E**) Immunofluorescence staining of un-injected oocytes (control) or oocytes injected with *Trpm7* (TRPM7) or *Trpm7* and *Arl15* cRNAs (TRPM7+ ARL15, ratio 10:1) using *anti*-M7d antibody and *anti*-mouse antibody conjugated with Alexa Fluor 488. Confocal images of Alexa Fluor 488 fluorescence (Alexa488) and overlays of Alexa488 with differential interference contrast images (overlay) are depicted for two independent oocytes per image; scale bars, 50 µm. The diagrams depict fluorescence intensity acquired along the green bars shown in *overlay* images. The stars indicate the cell surface of two oocytes. Typical examples of two independent experiments (n = 10 oocytes) are shown.

The online version of this article includes the following figure supplement(s) for figure 3:

**Figure supplement 1.** Two-electrode voltage clamp (TEVC) measurements of capsaicin-induced TRPV1 currents in *Xenopus* oocytes.

**Figure supplement 2.** Heterologous expression of transient receptor potential melastatin-subfamily member 7 (TRPM7), ARL8A, and ADP-ribosylation factor-like protein 15 (ARL15) in *Xenopus* oocytes.

**Figure supplement 3.** Assessment of the importance of the transient receptor potential melastatin-subfamily member 7 (TRPM7) kinase activity for the functional interplay between ADP-ribosylation factor-like protein 15 (ARL15) and TRPM7 by two-electrode voltage clamp (TEVC) measurements.

**Figure supplement 4.** Impact of ADP-ribosylation factor-like protein 15 (ARL15) on endogenous transient receptor potential melastatin-subfamily member 7 (TRPM7) currents in HEK293 cells.

Next, we studied whether heterologous expression in mammalian cells would allow uncovering any functional effects of CNNM3 on TRPM7. We transiently transfected HEK293 cells with *Trpm7* and *Cnnm3* plasmid cDNAs (ratio 2:1) and performed patch-clamp measurements (*Figure 4—figure supplement 1*). TRPM7 currents were induced using the standard divalent cation-free internal solution and an external buffer containing 1 mM $CaCl_2$ and 2 mM $MgCl_2$. When currents were developed, cells were exposed to mannitol-based saline containing 10 mM $Mg^{2+}$. In accord with previous publications (*Ferioli et al., 2017*), the perfusion of TRPM7-expressing cells with 10 mM $Mg^{2+}$ led to a significant reduction of outward currents accompanied by a relatively modest decrease of inward currents (*Figure 4—figure supplement 1*). Corresponding experiments with cells co-expressing TRPM7 and CNNM3 showed similar results (*Figure 4—figure supplement 1*), compatible with a TRPM7 $Mg^{2+}$ permeability unaltered by co-expression of CNNM3, regardless of the heterologous expression system.

Previously, we found that TRPM7 controls the uptake of $Mg^{2+}$ to maintain the cellular content of this mineral in resting cells (*Mittermeier et al., 2019*). To investigate whether CNNM3 modulates TRPM7-dependent $Mg^{2+}$ uptake, we employed inductively coupled plasma mass spectrometry (ICP-MS) to compare total amounts of magnesium in $TRPM7^{-/-}$ HEK293 cells transfected with *Trpm7*, *Cnnm3*, or *Trpm7* plus *Cnnm3* cDNAs (*Figure 4—figure supplement 2*). Next, we normalised the levels of magnesium to cellular sulphur (a biomarker for the total protein content) and observed that transient expression of TRPM7 increased the cellular Mg content, whereas expression of CNNM3 did not change this parameter (*Figure 4—figure supplement 2*). Importantly, we found that co-expression of TRPM7 with CNNM3 did not impact the ability of TRPM7 to regulate the cellular content of $Mg^{2+}$ (*Figure 4—figure supplement 2*). Hence, different experimental approaches did not reveal significant effects of CNNM3 on TRPM7 channel activity.

Since TRPM7 contains a C-terminal kinase domain, we studied whether CNNM3 might modulate the TRPM7 kinase moiety (*Figure 5* and *Figure 5—figure supplement 1*). To asses the activity of the TRPM7 kinase, we relied on the *anti*-(p)Ser1511 M7 antibody, which specifically recognises the known autophosphorylation site (Ser1511) of mouse TRPM7 (*Romagnani et al., 2017*). To verify that autophosphorylation of Ser1511 is dynamic, and changes of the TRPM7 kinase activity could therefore be visualised by the anti-(p)Ser1511 M7 antibody we treated HEK293 cells transiently overexpressing TRPM7 with TG100-115, a drug-like TRPM7 kinase inhibitor (*Song et al., 2017*). We observed that the exposure of living cells to TG100-115 led to suppression of (p)Ser1511 TRPM7 immunoreactivity in a dose-dependent fashion (*Figure 5—figure supplement 1A*). Moreover, the inhibitory effect of TG100-115 was time-dependent and could be detected 10 min after application of TG100-115

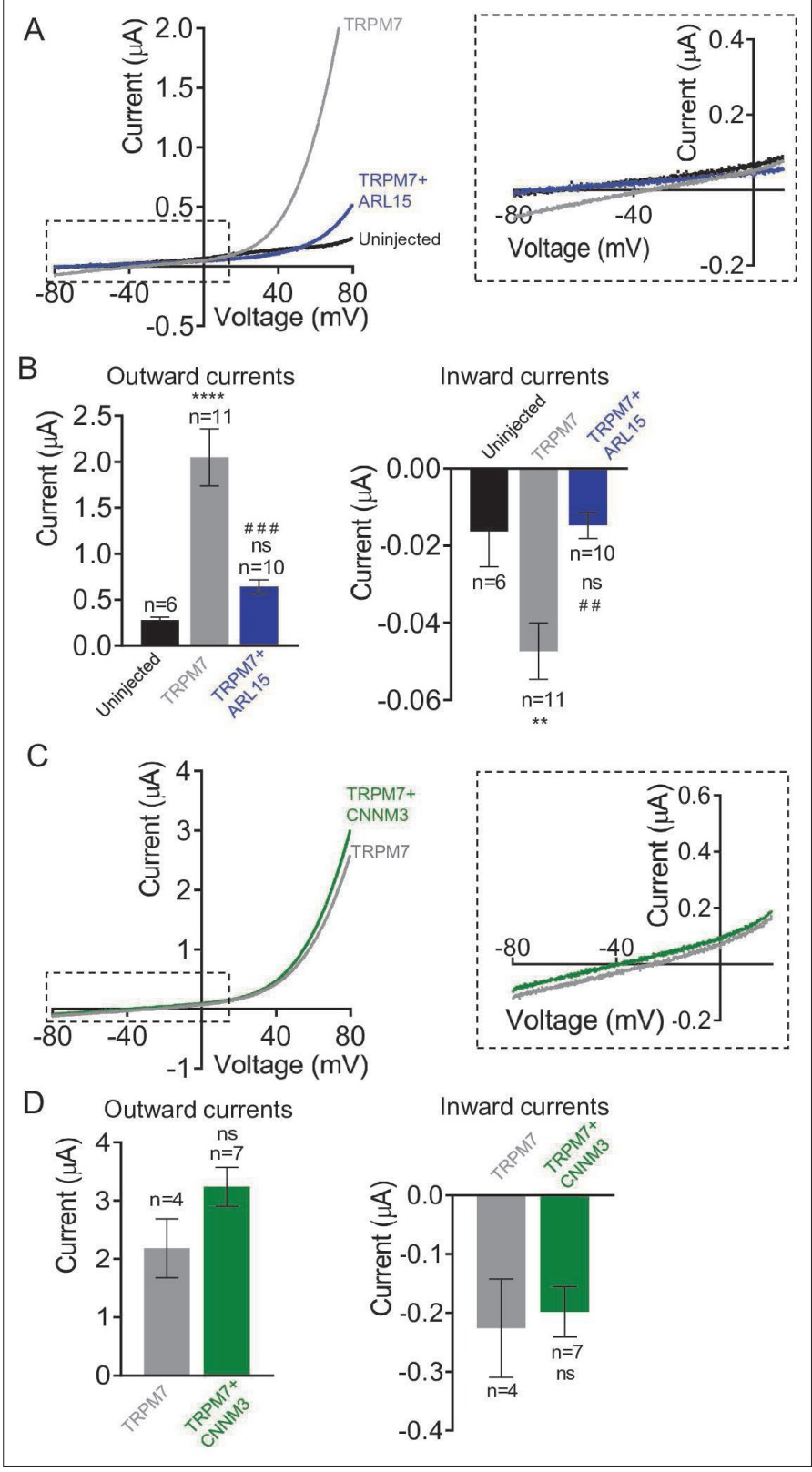

**Figure 4.** Effects of ADP-ribosylation factor-like protein 15 (ARL15) and CNNM3 on $Mg^{2+}$ currents of the transient receptor potential melastatin-subfamily member 7 (TRPM7) channel expressed in *Xenopus* oocytes. TEVC measurements were performed using the external ND96 solution containing 3 mM Mg2+and no other divalent cations. (A, B) Assessment of oocytes expressing TRPM7 or co-expressing TRPM7 with ARL15 (cRNA ratio

*Figure 4 continued on next page*

*Figure 4 continued*

10:1). (A) Representative I-V relationships ofTRPM7 currents. The dashed box in Left panelindicates the area of inward currents enlarged in the Right panel. (B) Current amplitudes (mean ± SEM) at+80 mV (Outward currents) and at -80 mV (Inward currents) in measurements from (A). Two independent batches of injected oocytes (n=6-11) were examined. ns, not 36significant; ** P < 0.01, **** P < 0.0001 significant to the Uninjected group (ANOVA). # # P < 0.01, # # # P < 0.001 significant to the TRPM7 group (ANOVA). (C, D) Examination of oocytes expressing TRPM7 or co-expressing TRPM7 with CNNM3 (cRNA ratio 2:1). Data were produced and analyzed as explained in (A, B). Two independent batches of injected oocytes (n=4-7) were examined. ns, not significant (two-tailed t-test).

The online version of this article includes the following figure supplement(s) for figure 4:

**Figure supplement 1.** Heterologous expression of transient receptor potential melastatin-subfamily member 7 (TRPM7) and CNNM3 in HEK293T cells.

**Figure supplement 2.** Assessment of total magnesium levels in *TRPM7$^{-/-}$* HEK293T cells transiently transfected with *Trpm7* and *Cnnm3* plasmid cDNAs.

(*Figure 5—figure supplement 1B*). Furthermore, we found that wash-out of TG100-115 by fresh cell culture medium caused a fast recovery of the (p)Ser1511 TRPM7 signal (*Figure 5—figure supplement 1C*). Hence, detection of (p)Ser1511 TRPM7 levels seems a reliable means to monitor the TRPM7 kinase activity. Accordingly, we investigated whether co-expression of ARL15 could modulate TRPM7 kinase activity and found no changes in (p)Ser1511 TRPM7 immunoreactivity (*Figure 5*). Co-expression of CNNM3 however caused a significant reduction of the (p)Ser1511 TRPM7 signal (*Figure 5*), suggesting that CNNM3 functions as a negative regulator of the TRPM7 kinase.

## Identification of new phosphorylation sites in the TRPM7 protein

In addition to subunit assembly, the MS data provided further insight into the post-translational modification(s) of the TRPM7 protein. Thus, TRPM7 purified either from rodent brain or from transfected HEK293 cells showed very similar patterns of serine and threonine phosphorylation, reflected by matching MS/MS spectra of peptides harbouring phosphorylation sites (*Figure 6A*, *Figure 6—figure supplement 1*, *Supplementary file 2* to *Figure 6*). Out of the nine shared phospho-sites, four have not been reported for TRPM7 in native tissue before (S1300, S1360, T1466, and S1567; *Supplementary file 2* to *Figure 6*). An additional 26 phosphorylated serine and threonine residues could be assigned to TRPM7 isolated from HEK 293 cells, presumably based on the higher amounts of TRPM7 available for analysis from heterologous (over)-expression material; 22 of these 26 sites match sites previously reported for TRPM7 endogenously or heterologously expressed in cell lines, and four sites were newly detected (S1208, S1480, S1496, S1853; *Supplementary file 2* to *Figure 6*). Most of the identified phosphorylation sites were found to cluster within the C-terminal cytoplasmic domain of TRPM7.

Finally, we asked whether measuring TRPM7 channel activity by TEVC would reveal any functional consequences of TRPM7 phosphorylation. We introduced phosphomimetic mutations in a subset of identified phospho-sites (S1208D, S1360D, S1480D, S1496D, and S1567D) and found that three TRPM7 mutants (S1208D, S1496D, and S1567D) displayed enhanced current amplitudes (*Figure 6B and C*), whereas their expression levels were similar to WT TRPM7 (*Figure 6D*). These findings suggest that phosphorylation of TRPM7 may represent a new regulatory mechanism reminiscent of the situation with TRPM8 (*Rivera et al., 2021*). To substantiate this notion further, it will be interesting to carry out a systematic functional analysis of the surprisingly extensive phosphorylation profile of TRPM7 (*Figure 6*).

## Discussion

In the present study, we investigated the molecular appearance and subunit composition of TRPM7 as present in the cell membrane(s) of the rodent brain. We show that TRPM7 forms macromolecular complexes by assembling with CNNM proteins 1-4 and ARL15. Moreover, functional expression in heterologous expression systems showed that ARL15 strongly affects TRPM7 channel function, while CNNM3 appears to act as a negative regulator of TRPM7 kinase activity.

BN-PAGE of membrane fractions isolated from rodent brain and cultured HEK 293 cells identified endogenous TRPM7 in high ~1.2 MDa molecular weight complexes exceeding the calculated

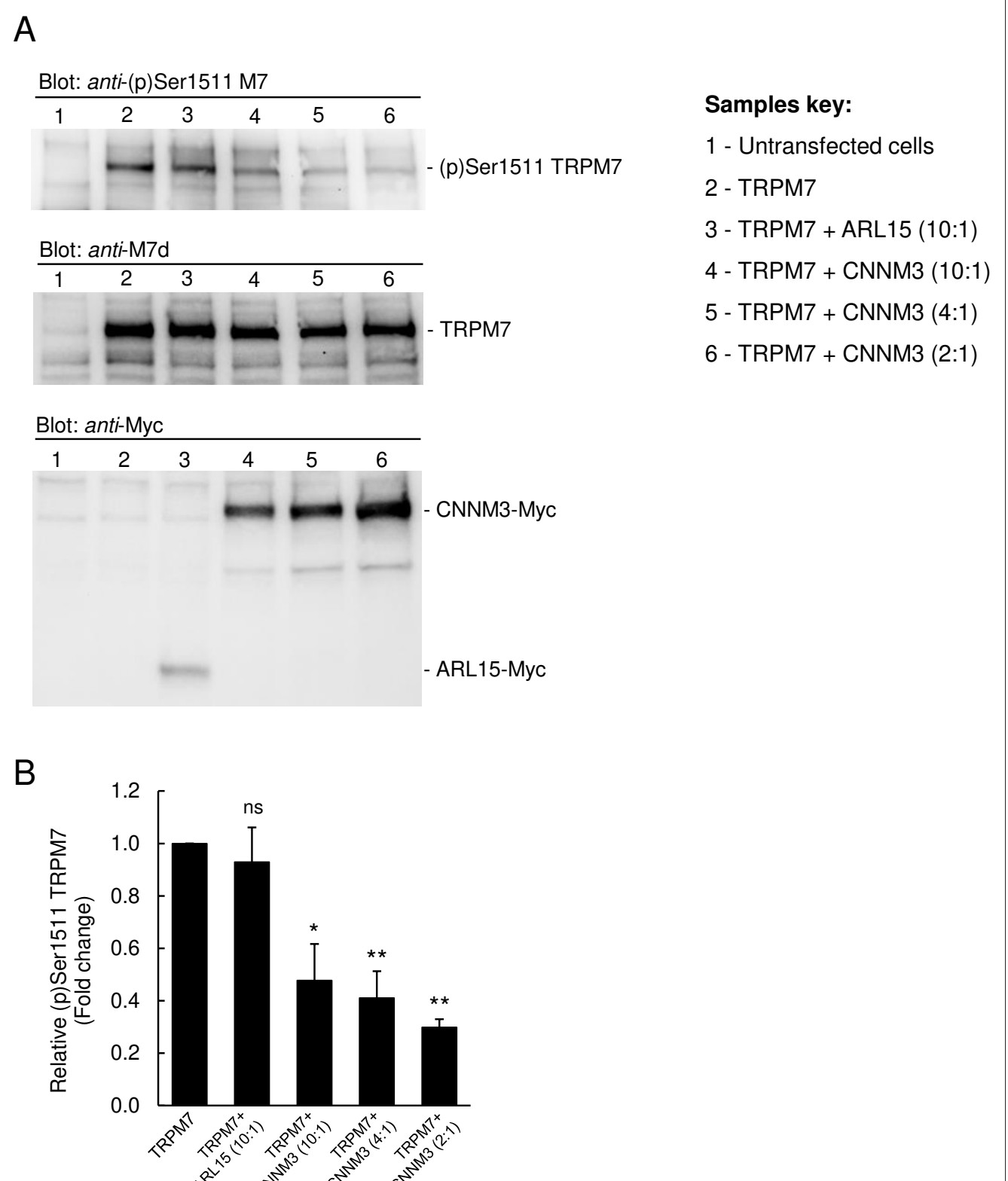

**Figure 5.** Impact of ADP-ribosylation factor-like protein 15 (ARL15) and CNNM3 on transient receptor potential melastatin-subfamily member 7 (TRPM7) autophosphorylation at Ser1511. (**A**) HEK293 cells were transiently transfected with *Trpm7*, co-transfected with *Trpm7* and *Arl15*, or with *Trpm7* and different amounts of *Cnnm3* plasmid cDNAs.  Twenty-four hours after transfection, cell lysates were examined using an *anti*-(p)Ser1511 M7 antibody (*upper panel*). After a stripping step, the blot was probed with *anti*-M7d (*middle panel*) and anti-*Myc* antibodies (*lower panel*) to detect total levels of

*Figure 5 continued on next page*

Figure 5 continued

TRPM7, ARL15-Myc, and CNNM3-Myc, respectively. Representative results are shown from three independent experiments. (**B**) Quantification of (p)Ser1511 TRPM7 levels in Western blot experiments (n = 3) shown in (**A**). A relative band density for each sample was obtained by dividing the (p)Ser1511 signal (*upper panel*) by the corresponding *anti*-M7d value (*middle panel*). The relative density of *Sample 2* (TRPM7) was set as a 1.0 to calculate changes in (p)Ser1511 TRPM7 (mean ± standard error of the mean [SEM]) caused by co-transfection of *Arl15* or *Cnnm3* as outlined in the bar graph. ns, not significant; *p ≤ 0.05, **p ≤ 0.01 significant to the control (ANOVA).

The online version of this article includes the following figure supplement(s) for figure 5:

**Figure supplement 1.** Effects of TG100-115 on transient receptor potential melastatin-subfamily member 7 (TRPM7) autophosphorylation.

molecular mass of TRPM7 tetramers (~850 kDa) and suggesting that the TRPM7 channel kinase is predominantly embedded in a large macromolecular complex. Compared to other native TRP channels, such as TRPC4, TRPM3, and TRPV2, the expression level of TRPM7 was found to be up to three orders of magnitude lower, thus classifying TRPM7 as a very low-abundant protein in the rodent brain and indicating that comprehensive determination of the TRPM7 complexome is technically challenging. The unbiased ME-AP approach paired with stringent negative controls nevertheless allowed for the identification of high-confidence interaction partners based on their specific and consistent co-purification with TRPM7. Consequently, five proteins were found to assemble with native TRPM7, including four members of the *CNNM* gene family encoding putative Mg$^{2+}$ transporters CNNM1-4 and a small G-protein ARL15. The fact that we did not detect all the interactors seen in mouse brain also in APs from rat brain is most likely due to the low abundance of endogenous TRPM7 (~50% less TRPM7 compared to APs from mouse brain). The interaction of TRPM7 with ARL15 and CNNM proteins was successfully confirmed in heterologous expression experiments. We also noted that previous proteome-wide interactome screens in cultured cells suggested an association of ARL15 with TRPM7 (*Huttlin et al., 2017*; *Huttlin et al., 2021*), in line with our results.

To obtain first insight into a possible functional impact of ARL15 and CNNM3, the most prominent interaction partners of TRPM7 in our experimental settings, we measured the channel activity of TRPM7 expressed in *Xenopus* oocytes and HEK293 cells. We found that co-expression of TRPM7 with CNNM3 did not lead to significant changes in TRPM7 currents applying a broad range of experimental conditions. Consistently, we observed that the ability of TRPM7 to increase cellular Mg levels was not affected by CNNM3. However, CNNM3 appears to act as a negative regulator of the TRPM7 kinase activity, resembling the action of the drug-like kinase inhibitor TG100-115. Collectively, these results suggest that CNNM3 may represent the first known protein acting as a physiological modulator of the TRPM7 kinase activity.

In contrast to CNNM3, co-expression of TRPM7 with ARL15 in oocytes, but not with the closely related small G-protein ARL8A, caused robust suppression of TRPM7 currents regardless of the experimental conditions applied. Of note, transient expression of ARL15 in HEK 293 cells resulted in inhibition of endogenous TRPM7 currents, reinforcing our conclusion that ARL15 acts as a potent and specific negative regulator of the TRPM7 channel.

The *CNNM* (Cyclin M; CorC) gene family encodes highly conserved metal transporter proteins identified in all branches of living organisms, ranging from prokaryotes to humans (*Funato and Miki, 2019*; *Giménez-Mascarell et al., 2019*). There are four family members in mammals, CNNM1-4, widely expressed in the body and abundantly present in the brain (*Funato and Miki, 2019*; *Giménez-Mascarell et al., 2019*). The genetic inactivation of *Cnnm4* in mice leads to systemic Mg$^{2+}$ deficiency (*Yamazaki et al., 2013*). In humans, point mutations in *CNNM2* cause hypomagnesemia (*Stuiver et al., 2011*), while mutations in *CNNM4* are associated with Jalili syndrome (*Parry et al., 2009*). Functional expression studies proposed that CNNMs operate as Na$^+$/Mg$^{2+}$ exchangers responsible for the efflux of cytosolic Mg$^{2+}$ from the cell (*Funato and Miki, 2019*; *Giménez-Mascarell et al., 2019*). In contrast to this view, other investigators proposed that CNNM proteins indirectly regulate the influx of Mg$^{2+}$ into the cell (*Arjona and de Baaij, 2018*). Recently resolved crystal structures of two prokaryotic CNNM-like proteins revealed that CNNMs form dimers and that each monomer contains three transmembrane helices harbouring Mg$^{2+}$ and Na$^+$ binding sites consistent with the suggested Na$^+$-coupled Mg$^{2+}$ transport function of CNNMs (*Huang et al., 2021*; *Chen et al., 2021*). While the majority of CNNM proteins in a cell is not bound to TRPM7, the direct association identified in this study suggests a new concept implying that two transporting mechanisms, TRPM7-mediated influx of divalent cations (Zn$^{2+}$, Mg$^{2+}$, and Ca$^{2+}$) and CNNM-dependent Na$^+$/Mg$^{2+}$ exchange, can be physically coupled under

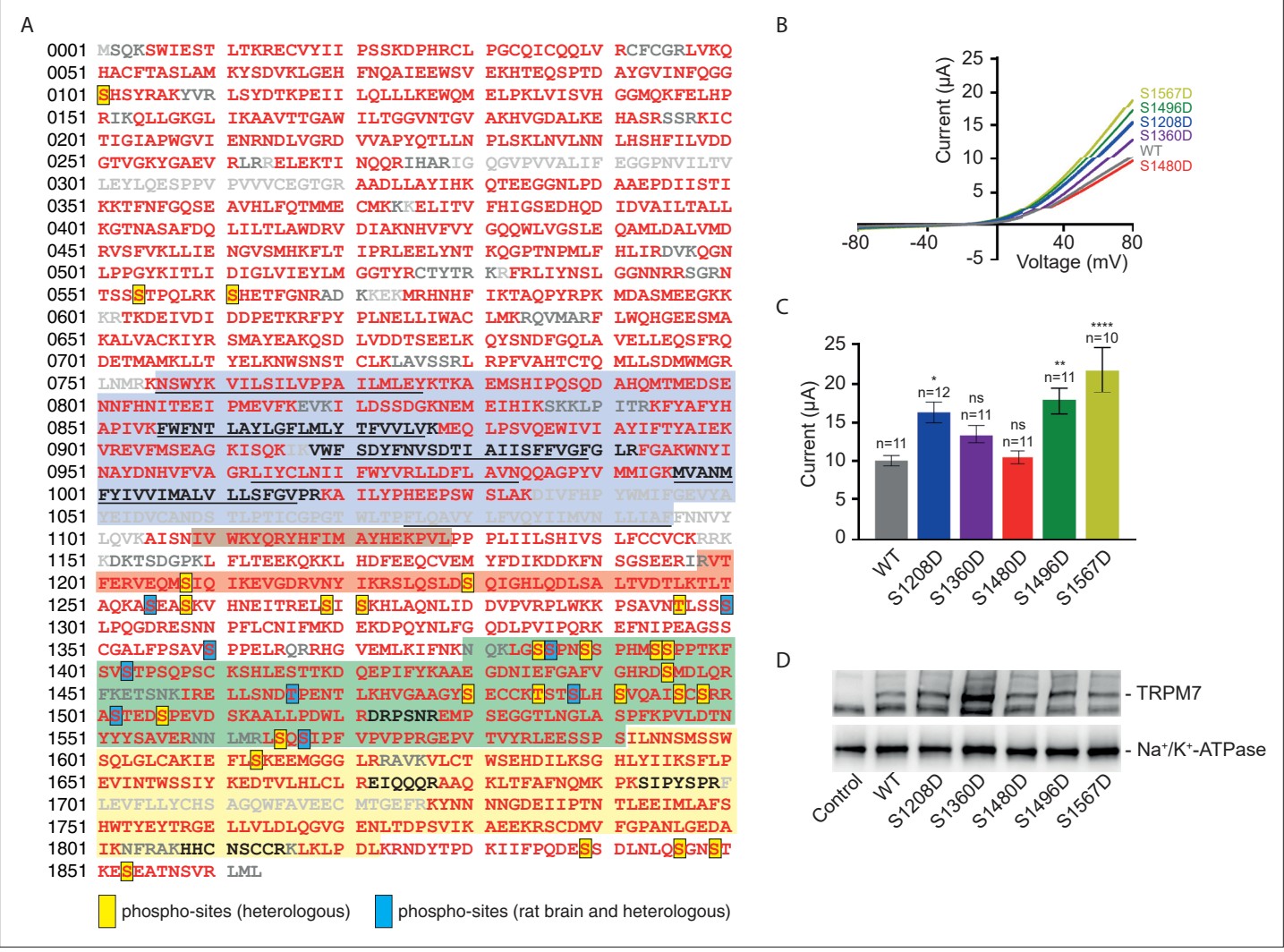

**Figure 6.** Identification of transient receptor potential melastatin-subfamily member 7 (TRPM7) phospho-sites and functional assessment of phosphomimetic TRPM7 mutants. (**A**) Coverage of the primary sequence of TRPM7 and phosphorylation sites as identified by mass spectrometry (MS) analyses of affinity purifications (APs) from transfected HEK293 cells and rodent brain. Peptides identified by MS are in red; those accessible to but not identified in tandem mass spectrometry (MS/MS) analyses are in black, and peptides not accessible to the MS/MS analyses used are given in grey. Blue boxes indicate phospho-sites identified in the brain and transfected HEK293 cells; those uniquely seen in heterologous expressions are boxed in yellow. Colour coding of hallmark domains is as in *Figure 1A*; S1-S6 helices of TRPM7 are underlined. (**B, C**) Two-electrode voltage clamp (TEVC) measurements of phosphomimetic TRPM7 mutants performed and analysed as explained in *Figure 3A*. (**B**) Representative current-voltage (I-V) relationships of TRPM7 currents measured in oocytes expressing WT and mutant variants of TRPM7, as indicated. (**C**) Current amplitudes (mean ± standard error of the mean [SEM]) at +80 mV of measurements shown in (**B**). Two independent batches of injected oocytes (n = 10–12) were examined. ns, not significant; *p ≤ 0.05, **p ≤ 0.01, ****p ≤ 0.0001 (ANOVA). (**D**) Western blot analysis of TRPM7 variants with phosphomimetic mutations expressed in *Xenopus* oocytes. Lysates of un-injected oocytes (control) or oocytes injected with WT and indicated mutant variants of *Trpm7* cRNAs were examined using the *anti*-M7d antibody. The *anti*-Na+/K+ ATPase antibody was used for loading controls. Representative results are shown for three independent experiments.

The online version of this article includes the following figure supplement(s) for figure 6:

**Figure supplement 1.** Tandem mass spectrometry (MS/MS) spectra illustrating phosphorylation of Ser1567 in transient receptor potential melastatin-subfamily member 7 (TRPM7) from both brain (upper panel) and culture cells (lower panel).

native conditions, thus, warranting future studies to examine the exact functional interplay between the channel-kinase TRPM7 and CNNMs.

ARL15 is a member of the *ARF* gene family of small G-proteins (*Gillingham and Munro, 2007*). A common feature of ARFs is their ability to bind and regulate effector proteins in a GTP-dependent manner (*Gillingham and Munro, 2007*). GDP- and GTP-bound states of ARFs are controlled by GTPase-activating proteins (GAP) in conjunction with GTP exchange factors (GEF) (*Gillingham and*

*Munro, 2007*). The best-characterised ARFs are involved in membrane trafficking, phospholipid metabolism and remodelling of the cytoskeleton (*Gillingham and Munro, 2007*). While genome-wide association studies have linked ARL15 to systemic $Mg^{2+}$ homeostasis and energy metabolism in humans (*Corre et al., 2018*; *Richards et al., 2009*), the particular functional role and corresponding GAP, GEF, and effector proteins of ARL15 remain to be established. To this end, the strong effect of ARL15 in suppressing TRPM7 currents observed in our study may suggest that TRPM7 serves as a specific effector protein of ARL15. The significance of this modulatory effect for native TRPM7 in the rodent brain, however, remains to be shown.

In some TRPM7-APs from HEK293 cells, we detected TRPM6, a genetically related channel, and two proteins representing the gene family of phosphatase of regenerating liver 1 and 3 (also entitled protein tyrosine phosphatases type 4A1 and 3, TP4A1 and 3) (*Table 1*). The $Mg^{2+}$ transporter protein TRPM6 has been described to physically and functionally interact with TRPM7 (*Chubanov et al., 2004*; *Ferioli et al., 2017*; *Chubanov et al., 2016*). In the present study, TRPM6, even though detected, could not be consistently co-purified with multiple anti-TRPM7 antibodies, likely because TRPM6 is expressed at very low levels in the brain and HEK293 cells. Nevertheless, a previous study reporting that heterologously expressed ARL15 positively modulates TRPM6 (*Corre et al., 2018*) might suggest an overlap between the TRPM6 and TRPM7 interactomes.

Interestingly, a recent interactome screen based on lentiviral overexpression of tagged proteins in HEK293 and HTC116 cells revealed that TP4A1 and TP4A2 also interact with ARL15 and CNNMs (*Huttlin et al., 2017*; *Huttlin et al., 2021*). Furthermore, a hypothesis-driven search for interaction partners of CNNMs has shown that TP4A proteins assemble with CNNMs and that such interactions shape $Mg^{2+}$ efflux from cells (*Funato et al., 2014*; *Hardy et al., 2015*; *Gulerez et al., 2016*; *Kostantin et al., 2016*; *Zhang et al., 2017*; *Giménez-Mascarell et al., 2017*). These findings are commensurate with our observation that TP4A1 and TP4A3 could be found in TRPM7 APs at low amounts.

Hence, based on the present analysis of native TRPM7 complexes in conjunction with earlier interactome experiments and functional expression studies, it is tempting to speculate that TRPM7/ARL15/CNNMs/TP4As form a protein network orchestrating transport of divalent cations across the cell membrane.

## Materials and methods

**Key resources table**

| Reagent type (species) or resource | Designation | Source or reference | Identifiers | Additional information |
|---|---|---|---|---|
| Strain, strain background (*Mus musculus*) | C57BL/6 | Jackson Labs | JAX stock #000664 | Six weeks of age, equal numbers of male and female |
| Strain, strain background (*Ratus norvegicus*) | Wistar | Charles River | Strain code:003 | Six weeks of age, equal numbers of male and female |
| Strain, strain background (*Xenopus laevis*) | *Xenopus laevis* | NASCO | Cat#:LM00535 | |
| Cell line (human) | HEK293T | Sigma | Cat#:96121229; RRID:CVCL_2737 | |
| Cell line (human) | *TRPM7$^{-/-}$* HEK293T | DOI:10.1073/pnas.1707380114 | | |
| Cell line (human) | HEK293T-Rex cells stably expressing *TRPM7* | 10.1016 /s0092-8674(03)00556–7 | | |
| Antibody | Anti-HA (rat monoclonal) | Roche | Cat#:11867423001; RRID:AB_390918 | IP (3–15 µg per IP), WB (0.2 µg/ml) |
| Antibody | Anti-HA (mouse monoclonal) | Invitrogen | Cat#:26183; RRID:AB_2533056 | IP (3 µg per IP) |
| Antibody | Normal rabbit IgG | Millipore | Cat#:12–370; RRID:AB_145841 | IP (15 µg per IP) |

*Continued on next page*

*Continued*

| Reagent type (species) or resource | Designation | Source or reference | Identifiers | Additional information |
|---|---|---|---|---|
| Antibody | Anti-βArrestin 2 (mouse monoclonal) | Santy Cruz Biotechnology | Cat#:sc-13140; RRID:AB_626701 | WB (1 µg/ml) |
| Antibody | Anti-TRPC1 (rabbit polyclonal) | Other | 4921 | Gift from Veit Flockerzi Immunogen: N-terminus of mouse TRPC1, IP (15 µg per IP) |
| Antibody | Anti-TRPC3 (rabbit polyclonal) | Other | 1378 | Gift from Veit Flockerzi Immunogen: N-terminus of mouse TRPC3, IP (15 µg per IP) |
| Antibody | Anti-NMDAR1 (mouse monoclonal) | Millipore | Cat#:MAB1586; RRID:AB_11213180 | IP (15 µg per IP) |
| Antibody | Anti-LRRTM2 (rabbit polyclonal), | ProteinTech | Cat#:23094–1-AP; RRID:AB_2879209 | IP (15 µg per IP) |
| Antibody | Anti-DPP10 (mouse monoclonal) | Santa Cruz Biotechnology | sc-398108 | IP (15 µg per IP) |
| Antibody | Anti-RGS9 (goat polyclonal) | Santa Cruz Biotechnology | sc-8143; RRID:AB_655555 | IP (15 µg per IP) |
| Antibody | Anti-TRPM7 (mouse monoclonal) | Thermo Fisher Scientific | Cat#:MA5-27620; RRID:AB_2735401 | IP (15 µg per IP) |
| Antibody | Anti-TRPM7 (mouse monoclonal) | NeuroMab | Cat#:75–114; RRID:AB_2877498 | IP (15 µg per IP) |
| Antibody | Anti-(p)Ser1511 TRPM7 (mouse monoclonal) | DOI:10.1038/s41467-017-01960-z | | Affinity purified with peptide H2N-DSPEVD(p)SKAALLPC-NH2, WB (2 µg/ml) |
| Antibody | Anti-M7c (rabbit polyclonal) | DOI:10.1038/s41467-017-01960-z | | Affinity purified with peptide H2N-DSPEVDSKAALLPC-NH2, IP (15 µg per IP) |
| Antibody | Anti-M7d (2C7, mouse monoclonal) | This paper | | See 'Materials and methods, Antibodies', IP (15 µg per IP), WB (0.8 µg/ml), IF (1.6 µg/ml) |
| Antibody | Anti-TRPM7 (4F9, mouse monoclonal) | This paper | | See 'Materials and methods, Antibodies', WB (1.4 µg/ml) |
| Antibody | Anti-TRPM7 (rabbit polyclonal) | Millipore | Cat#:AB15562; RRID:AB_805460 | WB (1 µg/ml) |
| Antibody | Anti-Flag (mouse monoclonal) | Sigma | Cat#:F3165; RRID:AB_259529 | WB (1 µg/ml) |
| Antibody | Anti-βActin (rabbit polyclonal) | Bioss Inc | Cat#:bs-0061R; RRID:AB_10855480 | WB (0.5 µg/ml) |
| Antibody | *Anti*-rabbit IgG (goat polyclonal, HRP conjugate) | abcam | ab7090 | WB (1:30000) |
| Antibody | *Anti*-mouse IgG (goat polyclonal, HRP conjugate) | abcam | ab7068 | WB (1:10000) |
| Antibody | *Anti*-mouse IgG (horse polyclonal, HRP conjugate) | Cell Signaling Technology | Cat#:7076 | WB (1:1000) |
| Antibody | *Anti*-Na$^+$/K$^+$ ATPase (rabbit monoclonal, HRP conjugate) | Abcam | Cat#:ab185065 | WB (1:1000) |
| Antibody | *Anti*-Myc (mouse monoclonal, clone 9B11) | Cell Signaling Technology | Cat#:2276 | WB (1:1000) |

*Continued on next page*

*Continued*

| Reagent type (species) or resource | Designation | Source or reference | Identifiers | Additional information |
|---|---|---|---|---|
| Antibody | *Anti*-mouse IgG- Alexa Fluor 488 (goat IgG, Alexa Fluor 488 conjugate) | Thermo Fisher Scientific | Cat#:A11029 | 2 µg/ml |
| Recombinant DNA reagent | pT7-His$_6$-*Trpm7*-KD (plasmid) | This paper | | See 'Materials and methods, Antibodies' |
| Peptide, recombinant protein | His$_6$-TRPM7-KD (purified protein) | This paper | | See 'Materials and Methods, Antibodies' |
| Peptide, recombinant protein | TRPM7-KD (purified protein) | This paper | | See 'Materials and methods, Antibodies' |
| Recombinant DNA reagent | Mouse *Trpm7* cDNA in pIRES2-EGFP vector (plasmid) | DOI: https://doi.org/10.1038/s41598-017-08144-1 | | Expression in mammalian cells |
| Recombinant DNA reagent | Mouse *Trpm6* cDNA in pIRES2-EGFP vector (plasmid) | DOI: https://doi.org/10.1038/s41598-017-08144-1 | | Expression in mammalian cells |
| Recombinant DNA reagent | Human *TRPM6* cDNA in pIRES2-EGFP vector (plasmid) | DOI: https://doi.org/10.1038/s41598-017-08144-1 | | Expression in mammalian cells |
| Recombinant DNA reagent | Mouse *Trpm7* cDNA in pOG1 vector (plasmid) | DOI: 10.1073/pnas.0305252101 | | cRNA synthesis |
| Recombinant DNA reagent | Mouse *Trpm7*-Myc cDNA in pcDNA3.1/V5-His TA-TOPO vector (plasmid) | DOI: 10.1073/pnas.0305252101 | | Expression in mammalian cells |
| Recombinant DNA reagent | Mouse *Trpm7*-HA cDNA in pcDNA3.1/V5-His TA-TOPO vector (plasmid) | DOI:10.1073/pnas.0305252101 | | Expression in mammalian cells |
| Recombinant DNA reagent | Human *TRPV1*-His cDNA in pNKS2 vector (plasmid) | This paper | | See 'Materials and methods, Antibodies, Molecular biology' cRNA synthesis |
| Recombinant DNA reagent | Mouse *Cnnm1*-Myc-Flag in pCMV6-Entry (plasmid) | OriGene | Cat#:MR218318 | Expression in mammalian cells |
| Recombinant DNA reagent | Mouse *Cnnm2*-Myc-Flag in pCMV6-Entry (plasmid) | OriGene | Cat#:MR218370 | Expression in mammalian cells |
| Recombinant DNA reagent | Mouse *Cnnm3*-Myc-Flag in pCMV6-Entry (plasmid) | OriGene | Cat#:MR224758 | Expression in mammalian cells, cRNA synthesis |
| Recombinant DNA reagent | Mouse *Cnnm4*-Myc-Flag in pCMV6-Entry (plasmid) | OriGene | Cat#:MR215721 | Expression in mammalian cells |
| Recombinant DNA reagent | Mouse *Arl15*-Myc-Flag in pCMV6-Entry (plasmid) | OriGene | Cat#:MR218657 | Expression in mammalian cells, cRNA synthesis |
| Recombinant DNA reagent | Mouse *Arl8a*-Myc-Flag in pCMV6-Entry (plasmid) | OriGene | Cat#:MR201740 | Expression in mammalian cells, cRNA synthesis |
| Commercial assay or kit | Bio-Rad Protein Assay | Bio-Rad | Cat#:5000006 | Protein concentration determination |
| Chemical compound, drug | ComplexioLyte CL-47 | Logopharm | Cat#:CL-47–01 | Mild detergent buffer |
| Chemical compound, drug | ComplexioLyte CL-91 | Logopharm | Cat#:CL-91–01 | Detergent buffer with intermediate stringency |
| Chemical compound, drug | Trypsin, sequencing grade modified | Promega | Cat#:V5111 | |
| Chemical compound, drug | Leupeptin | Sigma | Cat#:L2884 | |

*Continued on next page*

*Continued*

| Reagent type (species) or resource | Designation | Source or reference | Identifiers | Additional information |
|---|---|---|---|---|
| Chemical compound, drug | Pepstatin A | Sigma | Cat#:P5318 | |
| Chemical compound, drug | Aprotinin | Roth | Cat#:A162.2 | |
| Chemical compound, drug | Phenylmethylsulfonyl fluoride | Roth | Cat#:6367.3 | |
| Chemical compound, drug | Iodoacetamide | Sigma | I6125 | |
| Chemical compound, drug | Aminocaproic acid | Roth | 3113.3 | |
| Chemical compound, drug | TG100-115 | Selleck Chemicals | Cat#:S1352 | |
| Software, algorithm | msconvert.exe | http://proteowizard.sourceforge.net/ | | |
| Software, algorithm | MaxQuant v1.6.3 | http://www.maxquant.org | | |
| Software, algorithm | Mascot 2.6 | Matrix Science, UK | | |
| Software, algorithm | CellWorks 5.5.1 | npi electronic https://www.npielectronic.com | | |
| Software, algorithm | ZEN 2.3 | Carl Zeiss https://www.zeiss.de | | |
| Software, algorithm | PatchMaster 2 × 90 | Harvard Bioscience https://www.heka.com | | |
| Software, algorithm | Studio Lite 4.0 | https://www.licor.com/bio/image-studio-lite | | |
| Other | Dynabeads Protein A | Invitrogen | Cat#:10002D | |
| Other | Dynabeads Protein G | Invitrogen | Cat#:10004D | |
| Other | Tissue embedding media | Leica | Cat#:14020108926 | Used to support gel slices during cryotomy |

## Antibodies

Antibodies used for APs were: *anti*-HA (11867423001, Roche) and *anti*-HA (26183, Invitrogen). TUC antibodies were: rabbit IgG (12–370, Millipore), *anti*-βArrestin 2 (sc-13140, Santa Cruz), *anti*-TRPC1 (4921, a gift from Veit Flockerzi), *anti*-Sac1 (ABFrontier), *anti*-TRPC3 (1378, a gift from Veit Flockerzi), *anti*-NMDAR1 (MAB1586, Sigma), *anti*-LRRTM2 (23094–1-AP, ProteinTech), *anti*-DPP10 (sc-398108, Santa Cruz), and *anti*-RGS9 (sc-8143, Santa Cruz).

*Anti*-TRPM7 mouse monoclonal antibody (anti-M7a, *Figure 1A*) *was purchased from Thermo Fisher Scientific* (clone S74-25, Product # MA5-27620). *Anti*-TRPM7 mouse monoclonal antibody (*anti*-M7b, *Figure 1A*) *was obtained from NeuroMab* (clone N74/25, Product # 75–114). Generation of a rabbit polyclonal *anti*-(p)Ser1511 TRPM7 antibody (*anti*-(p)Ser1511 M7, *Figure 5*) was described previously (*Romagnani et al., 2017*). Briefly, rabbits were immunised with a phosphorylated peptide H2N-DSPEVD(**p**)SKAALLPC-NH2 ((p)Ser1511 in mouse TRPM7) coupled via its C-terminal cysteine residue to keyhole limpet hemocyanin (Eurogentec, Belgium). The generated serum was subjected to two rounds of affinity chromatography: a fraction of the antibody was purified using the phosphorylated peptide. Next, an additional round of chromatography was conducted using a non-phosphorylated variant of the peptide (H2N-DSPEVDSKAALLPC-NH2). The latter fraction of antibody was used in AP experiments (*anti*-M7c antibody, *Figure 1A*).

Anti-TRPM7 2C7 mouse monoclonal antibody (*anti*-M7d, *Figure 1A*, *Figure 1—figure supplement 1*) was produced by Eurogentec (Belgium) as follows. The nucleotide sequence coding for His$_6$-tag followed by a cleavage site sequence for TEV protease and the amino acids 1501–1863 (kinase domain, KD) of mouse TRPM7 protein was synthesised in vitro and cloned into the prokaryotic expression

vector pT7. The resulting expression construct pT7-His$_6$-*Trpm7*-KD was verified by sequencing and transformed in *Escherichia coli* (BL21 DE3 pLysS). Next, the transformed *E. coli* strain was amplified in LB medium at 25°C; 1 mM IPTG was used for induction of the His$_6$-TRPM7-KD protein expression. The harvested cell pellet was disrupted by sonication. His$_6$-TRPM7-KD was identified in the soluble fraction of the lysate. His$_6$-TRPM7 was purified on an Ni Sepharose 6 Fast Flow column on an AKTA Avant 25 (GE Healthcare) using an imidazole gradient of 20–500 mM. The fraction containing His$_6$-TRPM7-KD was dialysed against a Tris buffer (0.5 mM EDTA, 1 mM DTT, and 50 mM Tris HCl pH 7.5). His$_6$-TRPM7-KD was subjected to TEV protease (New England Biolabs) digestion according to the manufacturer's instructions. Subsequently, non-digested His$_6$-TRPM7-KD and His$_6$-tagged fragments were removed using an Ni-Sepharose 6 Fast Flow column. The flow-through containing the cleaved TRPM7-KD was concentrated to 0.5 mg/ml in the Tris buffer and stored at –80°C. SDS-PAGE was used to verify the removal of the His$_6$-tag.

The standard mouse monoclonal antibody production program of Eurogentec (Belgium) was conducted to immunise four mice using the TRPM7-KD protein and to produce a library of hybridomas. ELISA and Western blot were used to screen the hybridomas and to perform a clonal selection. Two hybridoma clones, 2C7 and 4F9 (isotypes G1;K), were selected based on the antibody quality released in the culture medium. Both clones were propagated, and the corresponding cell culture media were collected for large-scale purification of the IgG fraction using Protein G affinity chromatography. The IgG fractions from 2C7 (0.8 mg/ml) and 4F9 (1.4 mg/ml) were dialysed in PBS and stored at –80°C. The specificity of the 2C7 and 4F9 IgGs (dilution 1:1000) was verified by Western blot analysis of HEK293T cells overexpressing the TRPM6 and TRPM7 proteins (*Figure 1—figure supplement 1*). The 2C7 antibody detected the mouse or human TRPM7, but not the mouse or human TRPM6 (*Figure 1—figure supplement 1*). In contrast, the 4F9 antibody detected only the mouse TRPM7 (*Figure 1—figure supplement 1*). Consequently, the 2C7 antibody (*anti*-M7d) was used in the present study.

Quantification of (p)Ser1511 TRPM7 and *anti*-M7d signals in *Figure 5* was performed using Image Studio Lite 4.0 software (https://www.licor.com/bio/image-studio-lite).

## Molecular biology

Mouse *Trpm7*, mouse *Trpm6*, and human *TRPM6* cDNA in pIRES2-EGFP vector were reported previously (*Chubanov et al., 2004*; *Ferioli et al., 2017*). cDNA encoding C-terminally His-tagged human *TRPV1* (NG_029716 *Hayes et al., 2000*) was cloned into the pNKS2 vector (*Gloor et al., 1995*) using standard restriction enzyme (BamHI/SmaI) cloning techniques. The mouse *Trpm7* cDNA in the pOG1 and mouse *Trpm7-Myc* and *Trpm7-HA* cDNA variants in pcDNA3.1/V5-His TA-TOPO vector were described earlier (*Chubanov et al., 2004*; *Ferioli et al., 2017*). Expression constructs encoding Myc-Flag-tagged (C-end) mouse *Cnnm1-4* and *Arl15*, and *Arl8A* cDNAs in the pCMV6-Entry expression vector were acquired from OriGene (MR218318 for *Cnnm1*, MR218370 for *Cnnm2*, MR224758 for *Cnnm3*, MR215721 for *Cnnm4*, MR218657 for *Arl15*, and MR201740 for *Arl8a*) and verified by sequencing. Point mutations in *Trpm7* were introduced using the QuikChange system (Thermo Fisher Scientific) according to the manufacturer's protocol and verified by sequencing (Eurofins, Germany).

## Biochemistry

Cell lines, transient transfection: HEK293T cells (Sigma, 96121229, identity confirmed by STR profiling) were cultured at 37°C, 5% CO$_2$ in Dulbecco's modified Eagle's high glucose GlutaMAX medium (Gibco) supplemented with 10% foetal calf serum (Gibco), 1% penicillin/streptomycin (Gibco) and 10 mM Hepes (Gibco). TRPM7$^{-/-}$ HEK293T cells (*Abiria et al., 2017*) were cultured as WT cells with an addition of 10 mM MgCl$_2$, 3 µg/ml blasticidin S (InvivoGen), and 0.5 µg/ml puromycin (Gibco) to the medium. HEK293T-Rex cells stably expressing the human *TRPM7* were maintained as reported previously (*Schmitz et al., 2003*). The cell lines were tested negative for mycoplasma before use.

WT HEK293T cells were transfected with polyethylenimine (Polysciences) using a DNA to polyethylenimine ratio of 1:2.5. For transfection of TRPM7$^{-/-}$ HEK293T cells (*Abiria et al., 2017*), plasmid cDNA was diluted to 30 µg/ml in Hank's balanced salt solution, precipitated by addition of 113 mM CaCl$_2$ (final concentration) and added to the cells in culture medium lacking blasticidin S, puromycin, and 10 mM MgCl$_2$. For transfection, *Trpm7*, *Arl15*, and *Cnnm3* plasmid DNAs were mixed at a ratio of 3:1:1.

Preparation of plasma membrane-enriched protein fractions: Freshly excised brains from 25 male and 25 female 6-week-old rats (Wistar, Charles River) or mice (C57BL/6, Jackson Labs) were homogenised in homogenisation buffer (320 mM sucrose, 10 mM Tris/HCl pH 7.4, 1.5 mM $MgCl_2$, 1 mM EGTA and protease inhibitors leupeptin [Sigma], pepstatin A [Sigma], aprotinin [Roth] [1 µg/ml each], 1 mM phenylmethylsulfonyl fluoride [Roth], 1 mM iodoacetamide [Sigma]), particulates removed by centrifugation at 1080× *g* and homogenised material collected for 10 min at 200,000× *g*. After hypotonic lysis in 5 mM Tris/HCl pH 7.4 with protease inhibitors for 35 min on ice, the lysate was layered on top of a 0.5 and 1.3 M sucrose step gradient in 10 mM Tris/HCl pH 7.4, 1 mM EDTA/EGTA, and the plasma membrane-enriched fraction collected after centrifugation (45 min, 123,000× *g*) at the interface. Membranes were diluted in 20 mM Tris/HCl pH 7.4, collected by centrifugation (20 min, 200,000× *g*), and resuspended in 20 mM Tris/HCl pH7.4.

Cultured cells were harvested in phosphate buffer saline with protease inhibitors, collected by centrifugation (10 min, 500× *g*) and resuspended in homogenisation buffer. After sonication (2 × 5 pulses, duty 50, output 2 [Branson Sonifier 250]), membranes were pelleted for 20 min at 125,000× *g* and resuspended in 20 mM Tris/HCl pH 7.4. Protein concentration was determined with the Bio-Rad Protein Assay kit according to the manufacturer's instructions.

Immunoprecipitation: Membranes were resuspended in ComplexioLyte CL-47 or CL-91 solubilisation buffer (Logopharm) with added 1 mM EDTA/EGTA and protease inhibitors at a protein to detergent ratio of 1:8 and incubated for 30 min on ice. Solubilised protein was cleared by centrifugation (10 min, 125,000× *g*, 4°C) and incubated with antibodies cross-linked to Dynabeads (Invitrogen) by overhead rotation for 2 hr on ice. After two short washing steps with ComplexioLyte CL-47 dilution buffer (Logopharm), the captured protein was eluted in Laemmli buffer with dithiothreitol added after elution. Eluted proteins were separated by SDS-PAGE. For MS/MS analysis silver-stained (*Heukeshoven and Dernick, 1988*) protein lanes were cut-out, split at 50 kDa and pieces individually subjected to standard in-gel tryptic digestion (*Pandey and Mann, 2000*). For chemiluminescence detection, proteins were Western blotted onto PVDF membranes and probed with the following antibodies: *anti*-HA (11867423001, Roche), *anti*-Flag (F3165, Sigma), *anti*-βActin (bs-0061R, Bioss Inc).

BN-PAGE: Two-dimensional BN-PAGE/SDS-PAGE protein analysis was performed as described previously (*Schmidt et al., 2017*). Membrane protein fractions were solubilised in ComplexioLyte CL-47 as described above, salts exchanged for aminocaproic acid by centrifugation through a sucrose gradient, and samples loaded on non-denaturing 1–13% linear polyacrylamide gradient gels (anode buffer: 50 mM Bis-Tris, cathode buffer: 50 mM Tricine, 15 mM Bis-Tris, 0.02% Coomassie Blue G-250). For separation in the second dimension, individual gel lanes were isolated, equilibrated in 2× Laemmli buffer (10 min, 37°C), placed on top of SDS-PAGE gels and Western-probed using *anti*-TRPM7 (AB15562, Millipore).

## Complexome profiling

The size distribution of solubilised native TRPM7-associated complexes was investigated using the high-resolution csBN-MS technique detailed in *Faouzi et al., 2017*. Briefly, membranes isolated from adult mouse brain were solubilised with ComplexioLyte CL-47 (salt replaced by 750 mM aminocaproic acid), concentrated by ultracentrifugation into a 20%/50% sucrose cushion, supplied with 0.125% Coomassie G250 Blue and run overnight on a hyperbolic 1–13% polyacrylamide gel. The region of interest was excised from the lane, proteins fixed in 30% ethanol/15% acetic acid and the gel piece embedded in tissue embedding media (Leica). After careful mounting on a cryo-holder, 0.3 mm slices were harvested, rinsed, and subjected to in-gel tryptic digestion as described (*Faouzi et al., 2017*).

## Mass spectrometry

Tryptic digests (dried peptides) were dissolved in 0.5% (v/v) trifluoroacetic acid and loaded onto a C18 PepMap100 precolumn (300 µm i.d. ×5 mm; particle size 5 µm) with 0.05% (v/v) trifluoroacetic acid (5 min 20 µl/min) using split-free UltiMate 3000 RSLCnano HPLCs (Dionex/Thermo Scientific, Germany). Bound peptides were then eluted with an aqueous-organic gradient (eluent A: 0.5% (v/v) acetic acid; eluent B: 0.5% (v/v) acetic acid in 80% (v/v) acetonitrile; times referring to AP-MS/csBN-MS): 5 min 3% B, 60/120 min from 3% B to 30% B, 15 min from 30% B to 99% B or 20 min from 30% B to 50% B and 10 min from 50% B to 99% B, respectively, 5 min 99% B, 5 min from 99% B to 3% B, 15/10 min 3% B (flow rate 300 nl/min). Eluted peptides were separated in a SilicaTip emitter (i.d.

75 µm; tip 8 µm; New Objective, Littleton, MA) manually packed 11 cm (AP-MS) or 23 cm (csBN-MS) with ReproSil-Pur 120 ODS-3 (C18; particle size 3 µm; Dr Maisch HPLC, Germany) and electrosprayed (2.3 kV; transfer capillary temperature 250/300°C) in positive ion mode into an Orbitrap Elite (AP-MS) or a Q Exactive HF-X (csBN-MS) mass spectrometer (both Thermo Scientific, Germany). Instrument settings: maximum MS/MS injection time = 200/400 ms; dynamic exclusion duration = 30/60 s; minimum signal/intensity threshold = 2000/40,000 (counts), top 10/15 precursors fragmented; isolation width = 1.0/1.4 m/z.

Peak lists were extracted from fragment ion spectra using the 'msconvert.exe' tool (part of ProteoWizard [**Chambers et al., 2012**]; http://proteowizard.sourceforge.net/; v3.0.6906 for Orbitrap Elite and v3.0.11098 for Q Exactive HF-X; Mascot generic format with filter options 'peakPicking true 1-' and 'threshold count 500 most-intense'). Precursor m/z values were preliminarily searched with 50 ppm peptide mass tolerance, their mass offset corrected by the median m/z offset of all peptides assigned, and afterwards searched with 5 ppm mass tolerance against all mouse, rat, and human (mouse/rat brain samples) or only human (HEK293T cell samples) entries of the UniProtKB/Swiss-Prot database. Acetyl (protein N-term), carbamidomethyl (C), Gln-> pyro Glu (N-term Q), Glu-> pyro Glu (N-term E), oxidation (M), phospho (S, T, Y), and propionamide (C) were chosen as variable modifications, and fragment mass tolerance was set to ±0.8 Da (Orbitrap Elite data) or ±20 mmu (Q Exactive HF-X data). One missed tryptic cleavage was allowed. The expect value cut-off for peptide assignment was set to 0.5. Related identified proteins (subset or species homologs) were grouped using the name of the predominant member. Proteins either representing exogenous contaminations (e.g., keratins, trypsin, IgG chains) or identified by only one specific peptide were not considered.

Label-free quantification of proteins was carried out as described in **Bildl et al., 2012**; **Müller et al., 2016**. Peptide signal intensities (peak volumes, PVs) from FT full scans were determined, and offline mass calibrated using MaxQuant v1.6.3 (http://www.maxquant.org). Then, peptide PV elution times were pairwise aligned using LOESS regression (reference times dynamically calculated from the median peptide elution times overall aligned datasets). Finally, PVs were assigned to peptides based on their m/z and elution time (±1 min/2–3 ppm, as obtained directly or indirectly from MS/MS-based identification) using in-house developed software. PV tables were then used to calculate protein abundance ratios in AP versus control (**Figure 1C**), the abundance norm value (**Figure 1B**, lower right) as an estimate for molecular abundance (both described in **Schwenk et al., 2010**), and csBN-MS abundance profiles (**Figure 1B**, lower left) as detailed in **Müller et al., 2016**. The latter were smoothed by sliding, averaging over a window of 5. Slice numbers were converted to apparent complex molecular weights by the sigmoidal fitting of (log(MW)) versus slice number of the observed profile peak maximum of mitochondrial marker protein complexes (**Schägger and Pfeiffer, 2000**).

## Heterologous expression of TRPM7, CNNM3, ARL15, and ARL8A in *X. laevis* oocytes

TEVC measurements: *X. laevis* females were obtained from NASCO (Fort Atkinson, WI) and kept at the Core Facility Animal Models (CAM) of the Biomedical Center (BMC) of LMU Munich, Germany (Az:4.3.2–5682/LMU/BMC/CAM) in accordance with the EU Animal Welfare Act. To obtain oocytes, frogs were deeply anaesthetised in MS222 and killed by decapitation. Surgically extracted ovary lobes were dissociated by 2.5 hr incubation (RT) with gentle shaking in ND96 solution (96 mM NaCl, 2 mM KCl, 1 mM CaCl$_2$, 1 mM MgCl$_2$, 5 mM HEPES, pH 7.4) containing 2 mg/ml collagenase (Nordmark) and subsequently defolliculated by washing (15 min) with Ca$^{2+}$-free ND96. Stage V-VI oocytes were then selected and kept in ND96 containing 5 µg/ml gentamicin until further use.

TEVC measurements were performed as described previously (**Chubanov et al., 2004**) with a few modifications. Linearised cDNAs of *Trpm7* (in pOGI), *TRPV1* (in pNKS2), *Cnnm3*, *Arl8a*, and *Arl15* (all in pCMV6-Entry) were used for in vitro synthesis of cRNA (T7 or SP6 mMESSAGE mMACHINE transcription kits [Thermo Fisher Scientific]). In **Figure 3A**, oocytes were injected with 5 ng of *Trpm7* cRNA or co-injected with 2.5 ng of *Cnnm3* (2:1 ratio), 2.5 ng *Arl15* (2:1 ratio), and 2.5 ng of *Cnnm3* with 2.5 ng of *Arl15* cRNAs (2:1:1 ratio). In **Figure 3B**, oocytes were co-injected with 5 ng of *Trpm7* and 0.025–0.5 ng of *Arl15* cRNAs (200:1-10:1 ratio).

The injected oocytes were kept in ND96 solution, supplemented with 5 µg/ml gentamicin at 16°C. TEVC measurements were performed 3 days after injection at room temperature (RT) in Ca$^{2+}$/Mg$^{2+}$-free ND96 containing 3.0 mM BaCl$_2$ instead of CaCl$_2$ and MgCl$_2$ using a TURBO TEC-05X amplifier (npi

electronic) and CellWorks software (npi electronic). In some experiments, ND96 solution contained 3.0 mM $MgCl_2$ instead of 3.0 mM $BaCl_2$, as indicated in the corresponding figure legends. Oocytes were clamped at a holding potential of −60 mV, and 0.5 s ramps from −80 to +80 mV were applied at 6 s intervals. For statistical analysis, current amplitudes were extracted at −80 or +80 mV for individual oocytes, as indicated in the corresponding figure legends. Statistical significance (ANOVA) was calculated using GraphPad Prism 7.03.

Western blot: Oocytes *(n = 6 per group) were treated with* a lysis buffer (Pierce IP Lysis Buffer, Pierce) containing protease inhibitor and phosphatase inhibitor cocktails (Biotool), mixed (1:1) with 2× *Laemmli* buffer, heated at 70°C for 10 min, and cooled on ice. Samples were separated by SDS-PAGE (4–15% gradient Mini-PROTEAN, Bio-Rad) and electroblotted on nitrocellulose membranes (GE Healthcare Life Science). After blocking with 5% (w/v) non-fat dry milk in Tris-buffered saline with 0.1% Tween 20 (TBST). To probe for TRPM7 expression (*Figure 3D*), the upper part of the membrane was incubated with *anti*-M7d antibody (0.8 µg/ml) diluted in TBST with 5% (w/v) BSA, followed by washing in TBST, incubation with a horseradish peroxidase-coupled polyclonal horse anti-mouse IgG (#7076, Cell Signaling Technology; 1:1,000 in TBST with 5% (w/v) non-fat dry milk), and washing again in TBST. Blots were visualised using a luminescence imager (ChemiDoc Imaging System, Bio-Rad). The lower part of the membrane was developed using a horseradish peroxidase-coupled rabbit monoclonal *anti*-Na$^+$/K$^+$ ATPase antibody (ab185065, Abcam; 1:1000). To detect ARL15 (*Figure 3C*), the lower part of the membrane was incubated with a mouse *anti*-Myc antibody (clone 9B11, #2276, Cell Signaling Technology; 1:1000), and the upper part of the membrane was assessed by *anti*-Na$^+$/K$^+$ ATPase antibody.

Immunofluorescent staining: Oocytes *were* fixed in 4% (w/v) PFA (Electron Microscopy Sciences) in ND96 solution for 15 min at RT, followed by incubation in ice-cold methanol for 60 min at −18°C. After washing in ND96 (3×, RT), oocytes were incubated in ND96 containing 5% (w/v) BSA for 30 min at RT. *Anti*-M7d antibody (1.6 µg/ml in ND96 with 5% BSA) was applied overnight at 4°C. Afterwards, oocytes were washed in ND96 (3×, RT), and a goat *anti*-mouse IgG conjugated with Alexa Fluor 488 (Thermo Fisher Scientific; 2 µg/ml in ND96 with 5% BSA) was applied for 1 hr at RT. After washing in ND96 (3×, RT), differential interference contrast (DIC) and confocal images were obtained with a confocal laser scanning microscope LSM 880 AxioObserver (Carl Zeiss). We used a Plan-Apochromat 10×/0.45 objective, 488 nm excitation wavelengths and 493–630 nm filters. Acquired DIC and confocal images were analysed using the ZEN2.3 software (Carl Zeiss).

## Patch-clamp experiments with HEK293T cells

WT HEK293T cells were cultured using 3 cm dishes and Dulbecco's modified Eagle's medium (DMEM, high glucose; Merck) supplemented with 10% FBS, 100 µg/ml streptomycin, 100 U/ml penicillin (all from Thermo Fisher Scientific). Cells were maintained in a humidified cell culture incubator (Heraeus, Thermo Fisher Scientific) at 37°C and 5% $CO_2$. To investigate the effect of CNNM3 on the TRPM7 channel, cells were transiently transfected by 2 µg *Trpm7* (in pIRES2-EGFP) or 2 µg *Trpm7* plus 0.5 µg *Cnnm3* (in pCMV6-Entry) expression constructs using Lipofectamine 2000 reagent (Thermo Fisher Scientific). To examine the effects of ARL15 on endogenous TRPM7 currents, HEK293T cells were transfected by 1 µg WT *Arl15* (in pCMV6-Entry) and 0.1 µg *EGFP* cDNAs (in pcDNA3.1/V5-His TA-TOPO).

Patch-clamp measurements were conducted with EGFP-positive cells 18–22 hr after transfection, as reported previously (*Chubanov et al., 2004*; *Ferioli et al., 2017*), with minor modifications. Whole-cell currents were measured using an EPC10 patch-clamp amplifier and PatchMaster software (Harvard Bioscience). Voltages were corrected for a liquid junction potential of 10 mV. Currents were elicited by a ramp protocol from −100 to +100 mV over 50 ms acquired at 0.5 Hz and a holding potential of 0 mV. Inward and outward current amplitudes were extracted at −80 and +80 mV and were normalised to the cell size as pA/pF. Capacitance was measured using the automated capacitance cancellation function of EPC10. Patch pipettes were made of borosilicate glass (Science Products) and had resistance 2–3.5 MΩ. Unless stated otherwise, a standard extracellular solution contained (in mM): 140 NaCl, 2.8 KCl, 1 $CaCl_2$, 2 $MgCl_2$, 10 HEPES-NaOH, and 11 glucose (all from Sigma-Aldrich), pH 7.2. For assessing $Mg^{2+}$ currents, the extracellular solutions contained (in mM): 10 HEPES-NaOH, 260 mannitol, and 10 $MgCl_2$, pH 7.2. Solutions were adjusted to 290 mOsm using a Vapro 5520 osmometer (Wescor Inc). The standard divalent cation-free intracellular pipette solution contained (in mM): 120 Cs-glutamate, 8 NaCl, 10 Cs-EGTA, 5 Cs-EDTA, 10 HEPES-CsOH (all from Sigma-Aldrich), pH

7.2. Data are presented as means ± standard error of the mean (means ± SEM). Statistical comparisons (Prism 8.4.0) were made using one-way ANOVA or a two-tailed t-test, as indicated in the figure legends. Significance was accepted at $p \leq 0.05$.

## Determination of cellular Mg contents

The total content of Mg in *TRPM7*[-/-] HEK293T cells (*Abiria et al., 2017*) was determined by ICP-MS in ALS Scandinavia (Sweden) as reported previously (*Mittermeier et al., 2019*) with several modifications. The cells were cultured in DMEM (Merck) supplemented with 10% FBS, 100 µg/ml streptomycin, 100 U/ml penicillin, and 10 mM $MgCl_2$ (all from Thermo Fisher Scientific) in a humidified cell culture incubator (Heraeus, Thermo Fisher Scientific) at 37°C and 5% $CO_2$. To conduct ICP-MS experiments, *TRPM7*[-/-] HEK293T cells were plated in 10 cm² dishes at ~50% confluence in standard DMEM (without additional 10 mM $Mg^{2+}$) and transiently transfected with 20 µg *Trpm7*, 10 µg *Cnnm3*, or 20 µg *Trpm7* plus 10 µg *Cnnm3* plasmid cDNAs using Lipofectamine 2000 reagent (Thermo Fisher Scientific). After 24 hr, the cells were washed with serum-free DMEM, mechanically detached, and cell suspensions collected in 10 ml plastic tubes. After centrifugation (3 min, 1000 rpm), the medium was removed, and the cell pellet was resuspended in 5 ml PBS and passed to a fresh 10 ml tube. The cell suspension was centrifuged (3 min, 3500 rpm), the supernatant removed, and the cell pellet frozen at –20°C. Cell pellets were analysed by ICP-MS in ALS Scandinavia (Sweden). The experiment was repeated five times. Elementary Mg levels were normalised to elementary contents of sulphur (S) and represented as mean ± SEM. Data were compared by one-way ANOVA (Prism 8.4.0). Significance was accepted at $p \leq 0.05$.

## Acknowledgements

VC, TG, SZ, US, and BF were supported by the Deutsche Forschungsgemeinschaft (German Research Foundation, DFG), TRR 152 (P02, P14 and P15). BF and US were supported by the DFG under Germany's Excellence Strategy (CIBSS-EXC2189 project ID: 390939984) and Project-ID 403222702 – SFB 1381. AN was supported by the DFG Project-ID 335447717 – SFB 1328 (P15). TG and AN were supported by Research Training Group 2338 (DFG). We thank Veit Flockerzi for *anti*-TRPC1/3 antibodies, David Clapham for *TRPM7*[-/-] HEK293T cells, Carsten Schmitz for HEK293T-REx cells stably expressing *TRPM7*, and Ilia Rodushkin for the support in ICP-MS. We thank Joanna Zaisserer, Lisa Pleninger, Yves Haufe, Monika Haberland, and Anna Erbacher for their technical assistance.

## Additional information

### Funding

| Funder | Grant reference number | Author |
| --- | --- | --- |
| Deutsche Forschungsgemeinschaft | TRR 152 P15 | Vladimir Chubanov Thomas Gudermann |
| Deutsche Forschungsgemeinschaft | TRR 152 P02 | Bernd Fakler Uwe Schulte |
| Deutsche Forschungsgemeinschaft | SFB 1328 P15 | Annette Nicke |
| Deutsche Forschungsgemeinschaft | SFB 1381 | Bernd Fakler |
| Deutsche Forschungsgemeinschaft | Research Training Group 2338 | Thomas Gudermann Annette Nicke |
| Deutsche Forschungsgemeinschaft | TRR 152 P14 | Susanna Zierler |

The funders had no role in study design, data collection and interpretation, or the decision to submit the work for publication.

## Author contributions
Astrid Kollewe, Data curation, Formal analysis, Investigation, Visualization, Writing – original draft, Writing – review and editing; Vladimir Chubanov, Conceptualization, Data curation, Funding acquisition, Investigation, Resources, Supervision, Writing – original draft, Writing – review and editing; Fong Tsuen Tseung, Leonor Correia, Eva Schmidt, Anna Rössig, Catrin Swantje Müller, Wolfgang Bildl, Investigation; Susanna Zierler, Data curation, Formal analysis, Methodology, Visualization; Alexander Haupt, Formal analysis, Investigation, Writing – review and editing; Uwe Schulte, Data curation, Formal analysis, Funding acquisition, Writing – review and editing; Annette Nicke, Data curation, Formal analysis, Methodology, Resources, Writing – review and editing; Bernd Fakler, Conceptualization, Data curation, Funding acquisition, Project administration, Resources, Writing – original draft, Writing – review and editing; Thomas Gudermann, Conceptualization, Data curation, Funding acquisition, Project administration, Writing – original draft, Writing – review and editing

## Author ORCIDs
Vladimir Chubanov ![ORCID] http://orcid.org/0000-0002-6042-4193
Alexander Haupt ![ORCID] http://orcid.org/0000-0001-5647-5724
Annette Nicke ![ORCID] http://orcid.org/0000-0001-6798-505X
Bernd Fakler ![ORCID] http://orcid.org/0000-0001-7264-6423
Thomas Gudermann ![ORCID] http://orcid.org/0000-0002-0323-7965

## Decision letter and Author response
Decision letter https://doi.org/10.7554/eLife.68544.sa1
Author response https://doi.org/10.7554/eLife.68544.sa2

# Additional files

## Supplementary files
• Supplementary file 1. Numerical data for peak volumes, abundance norm values, relative abundance, and ratio distance values obtained through analysis of the mass spectrometry (MS) data.

• Supplementary file 2. Mass spectrometry (MS) spectra of phosphorylated transient receptor potential melastatin-subfamily member 7 (TRPM7), CNNM3, and CNNM4 peptides identified in affinity purifications (APs) from HEK293 and rodent brain.

• Supplementary file 3. Phosphorylation sites in transient receptor potential melastatin-subfamily member 7 (TRPM7), CNNM3, and CNNM4 identified in affinity purifications (APs) from transfected HEK293 cells and rodent brain. Excel file contains one worksheet: The phosphorylated residues of TRPM7, CNNM3, and CNNM4 identified by mass spectrometry (MS) in the present study are outlined in conjunction with previously published data (*Nguyen et al., 2019*; *Zhou et al., 2013*; *Cai et al., 2017*; *Huttlin et al., 2010*).

• Transparent reporting form

## Data availability
The mass spectrometry proteomics data have been deposited to the ProteomeXchange Consortium via the PRIDE partner repository with the dataset identifier PXD025279 and https://www.ebi.ac.uk/pride/archive/projects/PXD025279.

The following dataset was generated:

| Author(s) | Year | Dataset title | Dataset URL | Database and Identifier |
|---|---|---|---|---|
| Haupt A, Fakler B | 2021 | The molecular appearance of native TRPM7 channel complexes identified by high-resolution proteomics | https://www.ebi.ac.uk/pride/archive/projects/PXD025279 | PRIDE, PXD025279 |

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
