## [Editor Report]

This work will be interesting to people studying TRP family ion channels and more generally, cellular ion homeostasis. It is the first to identify interacting protein partners of the cation channel TRPM7, a key regulator of cellular Mg^2+^ and Zn^2+^ homeostasis, and reveals functional coupling between TRPM7, a putative magnesium transporter, and a small G protein.

---

## [Decision Letter]

**Decision letter after peer review:**

Thank you for submitting your article "The molecular appearance of native TRPM7 channel complexes identified by high-resolution proteomics" for consideration by *eLife*. Your article has been reviewed by 3 peer reviewers, including László Csanády as Reviewing Editor and Reviewer #1, and the evaluation has been overseen by Kenton Swartz as the Senior Editor. The following individuals involved in review of your submission have agreed to reveal their identity: Thomas Voets (Reviewer #2); Rajan Sah (Reviewer #3).

Essential revisions:

1. The authors' conclusions are based on MS analysis of co-immunoprecipitated endogenous proteins and on classical co-IP studies in cells overexpressing the target proteins. The conclusions would be strengthened by a more rigorous assessments of the demonstrated interactions through classical co-IP experiments between endogenous proteins in mammalian cells. This would eliminate the possibility of artifactual interactions that may occur with overexpression studies, and provide more direct support for the MS results.

2. The authors should provide an explanation of why ARL-15 co-IP's with TRPM7 only in the presence of CNNM3 (Figure 3), but suppresses TRPM7 currents in oocytes also when CNNM3 is not present. Is the inhibitory influence of ARL15 on TRPM7 currents indirect, not requiring the physical association of ARL-15 with the channel? If so, what is the relevance of the demonstrated physical interaction? One possible control experiment could be to co-express in oocytes TRPM7 with some other small GTPase (which does not co-IP with TRPM7).

3. Some additional experiments would be needed to address the functional consequence of the presence of the interaction partners on Mg^2+^ permeation and Mg^2+^-dependent regulation of TRPM7. For instance, patch-clamp experiments in HEK cells to determine relative Mg^2+^ permeability and to address whether the regulation by intracellular Mg^2+^/Mg^2+^-ATP is affected.

4. Some assessment of modulation of magnesium homeostasis by the TRPM7-CNNM interaction would be desirable, ideally in a mammalian cell.

*Reviewer #1:*

This is a concise manuscript which focuses on the identification of TRPM7 reaction partners of functional relevance in native tissues. Using native gel electrophoresis the authors show that TRPM7 is present in complexes with an apparent molecular weight (MW) that greatly exceeds the calculated MW of the homotetramer. They then find that upon affinity purification of TRPM7 from rodent brain or TRPM7-expressing HEK293 cells, using a variety of different epitopes, the magnesium transporters CNNM1-4 and the small G protein ARL15 consistently co-purify with the channel. Functional studies in *Xenopus oocytes* co-expressing various combinations of TRPM7, CNNM3, and ARL15 show that TRPM7 currents are unaffected by co-expression of CNNM3, but robustly suppressed by co-expression of ARL15. The latter effect is specific to TRPM7 (over TRPV1), scales with ARL15 expression levels, and is not mediated by a reduction in TRPM7 surface expression.

To my judgement the experiments are carefully performed and seem to support the key conclusions. Although the simple functional experiments do not offer mechanistic insight into the TRPM7-ARL15 interaction, this study opens up a completely novel avenue of research.

My only concern is an apparent logical contradiction between the findings presented in Figure 3 and those presented in Figures 1C and 4.

First, in Figure 3 the authors show that the interaction between TRPM7 and ARL15 depends on the presence of CNNM proteins: when CNNM3 is not co-expressed, TRPM7 does not co-precipitate ARL15 from HEK293 cells. On the other hand, in Figure 1C they show that CNNM3 is natively expressed in HEK293 cells, as it co-precipitates with heterologously expressed TRPM7.

Second, in *Xenopus oocytes*, co-expression of ARL15 alone is sufficient to suppress TRPM7 currents. Thus, either CNNM3 (or a homolog) is indeed required for the TRPM7-ARL15 interaction (as suggested by Figure 3), but is endogenously expressed in *Xenopus oocytes*, or the inhibitory influence of ARL15 on TRPM7 currents is indirect, and does not require the physical association of ARL-15 with the channel. If so, what is the relevance of the demonstrated physical interaction? The authors should comment on these issues.

*Reviewer #2:*

To understand the function and regulation of ion channels in the plasma membrane, knowledge of other proteins that are associated with channels in functional complexes is important. Here, Kollewe et al., used high resolution proteomics to assess the composition of native complexes containing TRPM7, a plasma membrane cation channel involved in cation transport, and containing a functional kinase in its C terminus.

Using affinity purification and mass spectrometry, they convincingly identify a specific interaction of TRPM7 with members of the CNNM family of putative magnesium transporters and with the small G protein ARL15. This interaction can be reproduced in heterologous expression system. Moreover, in *Xenopus oocytes*, increasing levels of ARL15 lead to a dose-dependent inhibition of TRPM7 currents.

Overall, this work represents an important next step in TRPM7 channel research, by systematically identifying the composition of the channel interactome, and how it affects channel functionality. The study can serve as a paradigm for similar studies on other (TRP) channels, which may yield a wealth of information on channel functional regulation.

At this point, the (patho)physiological relevance of the interactions with ARL15 and CNNM1-4 remain unclear. Further work is needed to pinpoint potential effects of these proteins on TRPM7-dependent (magnesium) ion transport or kinase activity.

Overall, this is meticulous study identifying potentially important interaction partners of the cation channel TRPM7. A weaker point of the paper is that the functional effects of the interaction partners are studied/described in a somewhat superficial manner:

1) The analysis of the currents measured in *Xenopus oocytes* only includes currents at +80 mV, which is far from the physiological range. It would be of interest to also provide additional data regarding the inward currents and, especially seen the interaction with CNNMs, the magnesium permeability of the complex.

2) TRPM7 is a channel and an enzyme. However, the authors only investigate the effects of the binding partners on channel activity, not on kinase activity. Would it be possible to include an assay investigating the effect of ARL15 or CNMMs on kinase activity? In any case, such a potential regulatory effect should be discussed.

Labeling in Figure 1B seems incorrect.

*Reviewer #3:*

Chubanov et al., applied multi-epitope affinity purification and high-resolution quantitative MS analysis and identified TRPM7 forming a ternary complex with CNNM proteins and ARL15. CNNM1-4 proteins, like TRPM7, are thought to be magnesium transporters, so this finding is very intriguing as it relates to the mechanisms by which TRPM7 and CNNM, either directly or indirectly regulate magnesium homeostasis. The authors confirm these interactions with overexpression co-IP experiments in HEK cells and then evaluate the effects of ARL15 expression on TRPM7 function in *Xenopus oocytes*. They reveal that ARL15 is a negative regulator of TRPM7 channel function while maintaining TRPM7 plasma membrane localization. These studies are overall well performed, and technically rigorous. These initial findings are also very interesting and potentially impactful as they may finally explain a longstanding controversy about how TRPM7 (and TRPM6) may regulate magnesium homeostasis.

1. The finding that TRPM7 forms a complex with CNNM proteins, another putative magnesium transporter, now with a recent cryo-EM structure that supports a role in magnesium transport, is a very important finding that deserves to be further explored in this paper. As the authors recognize, the connection of TRPM7 to CNNM proteins has been previously described as has the connection between CNNM proteins and ARL15, which reduces novelty. Moreover, this manuscript appears to end at these initial findings which would normally be the beginning of a more impactful and meaningful paper.

2. Relevant to the point above, how are CNNM protein expression and localization altered in TRPM7-/- cells? Could CNNM mistrafficking or impaired function account for the differences in magnesium homeostasis observed in TRPM7-/- cells?

3. Ion channel functional studies are performed in *Xenopus oocytes* and the rationale is well explained with respect to titrating expression levels, but why not perform these studies in HEK cells to confirm that this holds true in the more relevant mammalian cells?

4. Are there differences in intracellular magnesium upon expression of TRPM7 +/- CNNM proteins +/- ARL15 in overexpression studies in mammalian cells?

5. TRPM7 antibodies are notoriously challenging with respect to specificity. The authors described negative control experiments using TRPM7-/- cells. Where is the TRPM7 KO negative control data?

6. The co-IP experiments with overexpression of TRPM7 and then each of identified TRPM7 interactors, CNNM1-4, ARL15 is the least specific experiment possible for confirmation. Why not perform endogenous TRPM7 IP (since the Abs seem to work for IP) and then probe for endogenous, or even overexpressed CNNM and ARL15? Conversely, is it possible to IP endogenous, or overexpressed CNNM or ARL15 and detect endogenous TRPM7 on WB? These additional co-IP would strengthen this finding.

7. The new phosphosites identified are interesting, but what is their functional significance?

8. For the experiments in Figure 4, ARL15 expression is titrated based on injected mRNA, but it would also be advisable to show the actual levels of overexpressed protein by WB.

---

## [Author Response]

Essential revisions:1. The authors' conclusions are based on MS analysis of co-immunoprecipitated endogenous proteins and on classical co-IP studies in cells overexpressing the target proteins. The conclusions would be strengthened by a more rigorous assessments of the demonstrated interactions through classical co-IP experiments between endogenous proteins in mammalian cells. This would eliminate the possibility of artifactual interactions that may occur with overexpression studies, and provide more direct support for the MS results.

We are afraid that, unfortunately, a critical misunderstanding appears to have arisen concerning our central experimental approach. Rigorous analysis of native TRPM7 channels present in the rodent brain was the major goal of the present study. For this purpose, we isolated native TRPM7 channels by affinity-purifications (APs) with four different *anti-TRPM7* antibodies – in other words: We performed ‘classical APs’ that were evaluated by quantitative MS-analysis. In contrast to Western-blotting, MS-analysis is highly sensitive (dynamic range of four orders of magnitude), quantitative and unbiased (independent of antibody-specificities) – and thus the preferred method of choice for readout and evaluation of AP-based studies. In this sense it is important to note, that CNNMs and ARL15 were identified as interactors of native TRPM7 channels, which are, in fact, macro-molecular complexes assembled from the aforementioned proteins (Figure 1). Subsequent APs from un-transfected HEK-cells, as well as from HEK cells (over)-expressing TRPM7 protein confirmed these interactions by specific co-purification of endogenous CNNM and ARL15 proteins (Figure 1). For better clarity, we changed the labeling for the APs from HEK cells to highlight that endogenous proteins were purified in these experiments (Figure 1C, third panel).

2. The authors should provide an explanation of why ARL-15 co-IP's with TRPM7 only in the presence of CNNM3 (Figure 3), but suppresses TRPM7 currents in oocytes also when CNNM3 is not present. Is the inhibitory influence of ARL15 on TRPM7 currents indirect, not requiring the physical association of ARL-15 with the channel? If so, what is the relevance of the demonstrated physical interaction? One possible control experiment could be to co-express in oocytes TRPM7 with some other small GTPase (which does not co-IP with TRPM7).

Unfortunately, there seems to be a misinterpretation of Figure 3 (now Figure 2). Heterologously expressed TRPM7 co-assembles with ARL15 even in the absence of overexpressed CNNM3, although at lower efficiency (Figure 3 (now Figure 2), right panel, lane 5 versus lane 7). Currently, we cannot exclude that the effect of ARL15 expression observed in oocytes is based on co-assembly with *Xenopus*-endogenous CNNMs. Nevertheless, the effect is ARL15-specific.

We appreciate the advice of the reviewers about the importance of assessing other small G-proteins to demonstrate a specific effect of ARL15. We have made extensive additional efforts to address this issue. Using TEVC, we found that a genetically related protein, ARL8A, is not able to affect TRPM7 currents (new Figure 3—figure supplement 2). Next, we performed patch-clamp experiments with ARL15 and demonstrated that ARL15 can also inhibit endogenous TRPM7 currents (new Figure 3—figure supplement 4), showing that the inhibitory action of ARL15 is specific and does not depend on the expression system used. Finally, we examined the kinase-dead TRPM7 K1646R mutant to show that TRPM7 kinase activity is not required for the inhibitory action of ARL15 (new Figure 3—figure supplement 3C). Collectively, these data significantly extend the mechanistic insight into the functional interaction between ARL15 and TRPM7.

3. Some additional experiments would be needed to address the functional consequence of the presence of the interaction partners on Mg^2+^ permeation and Mg^2+^-dependent regulation of TRPM7. For instance, patch-clamp experiments in HEK cells to determine relative Mg^2+^ permeability and to address whether the regulation by intracellular Mg^2+^/Mg^2+^-ATP is affected.

We agree with the reviewers that the effects of ARL15 and CNNM3 on the Mg^2+^ permeability of TRPM7 will benefit the impact of the study. As suggested, we conducted patch-clamp experiments with HEK293 cells transfected with TRPM7 and CNNM3 (new Figure 4—figure supplement 1). We did not detect any impact of CNNM3 on TRPM7 channel characteristics.

In addition, we conducted TEVC experiments using an external solution solely containing 3 mM Mg^2+^ and no other divalent cations (new Figure 4). In line with our patch-clamp experiments, this approach did not show any functional effects of CNNM3 on TRPM7 currents. In contrast to CNNM3, investigating ARL15 coexpression in two expression systems (HEK293 cells and *Xenopus laevis* oocytes) using different experimental conditions consistently showed that ARL15 efficiently inhibits TRPM7 currents (new Figure 3—figure supplement 4, new Figure 4). Therefore, we conclude that ARL15 elicits a general suppression of TRPM7 channel activity and does not selectively impinge on specific channel characteristic such as Mg^2+^ permeability.

We would like to emphasize that during TEVC measurements of TRPM7 currents, cytosolic levels of Mg^2+^ and Mg-ATP were not manipulated, whereas patch-clamp measurements of whole-cell TRPM7 currents were performed with a divalent cation free internal solution that also contained the Mg^2+^ chelators EGTA and EDTA and therefore efficiently removed intracellular Mg^2+^ and Mg-ATP. Hence, the effects of ARL15 and CNNM3 on TRPM7 were examined in the presence and absence of Mg^2+^ (or Mg-ATP) with a similar outcome.

4. Some assessment of modulation of magnesium homeostasis by the TRPM7-CNNM interaction would be desirable, ideally in a mammalian cell.

As suggested, we employed inductively coupled plasma mass spectrometry (ICP-MS) to compare total magnesium concentrations in TRPM7^-/-^ HEK293T cells transfected with TRPM7, CNNM3 or TRPM7 plus CNNM3 cDNAs (new Figure 4—figure supplement 2). We found that, in accord with electrophysiological data, the ability of TRPM7 to regulate Mg^2+^ uptake was not altered by the presence of over-expressed CNNM3.

Reviewer #1:This is a concise manuscript which focuses on the identification of TRPM7 reaction partners of functional relevance in native tissues. Using native gel electrophoresis the authors show that TRPM7 is present in complexes with an apparent molecular weight (MW) that greatly exceeds the calculated MW of the homotetramer. They then find that upon affinity purification of TRPM7 from rodent brain or TRPM7-expressing HEK293 cells, using a variety of different epitopes, the magnesium transporters CNNM1-4 and the small G protein ARL15 consistently co-purify with the channel. Functional studies in *Xenopus oocytes* co-expressing various combinations of TRPM7, CNNM3, and ARL15 show that TRPM7 currents are unaffected by co-expression of CNNM3, but robustly suppressed by co-expression of ARL15. The latter effect is specific to TRPM7 (over TRPV1), scales with ARL15 expression levels, and is not mediated by a reduction in TRPM7 surface expression.To my judgement the experiments are carefully performed and seem to support the key conclusions. Although the simple functional experiments do not offer mechanistic insight into the TRPM7-ARL15 interaction, this study opens up a completely novel avenue of research.

We thank Reviewer 1 for such a favourable evaluation of our study.

My only concern is an apparent logical contradiction between the findings presented in Figure 3 and those presented in Figures 1C and 4.First, in Figure 3 the authors show that the interaction between TRPM7 and ARL15 depends on the presence of CNNM proteins: when CNNM3 is not co-expressed, TRPM7 does not co-precipitate ARL15 from HEK293 cells. On the other hand, in Figure 1C they show that CNNM3 is natively expressed in HEK293 cells, as it co-precipitates with heterologously expressed TRPM7.

The heterologous reconstitution experiments in HEK cells (Figure 3 (now Figure 2)) show that ARL15 is able to associate with TRPM7 also in the absence of overexpressed CNNM3, albeit at lower efficiency (Figure 3 (now Figure 2) right panel lane 5 versus lane 7). Whether or not this interaction requires recruitment of endogenous CNNM proteins could only be answered with a cell line deprived of all 4 CNNMs, which, to our knowledge, is currently not available.

Second, in *Xenopus oocytes*, co-expression of ARL15 alone is sufficient to suppress TRPM7 currents. Thus, either CNNM3 (or a homolog) is indeed required for the TRPM7-ARL15 interaction (as suggested by Figure 3), but is endogenously expressed in *Xenopus oocytes*, or the inhibitory influence of ARL15 on TRPM7 currents is indirect, and does not require the physical association of ARL-15 with the channel. If so, what is the relevance of the demonstrated physical interaction? The authors should comment on these issues.

Unfortunately, Figure 3 (now Figure 2) appears to be misinterpreted. Heterologously expressed TRPM7 co-assembles with ARL15 even in the absence of overexpressed CNNM3, although at lower efficiency (Figure 3 (now Figure 2), right panel, lane 5 versus lane 7). Currently, we cannot exclude that the effect of ARL15-expression observed in oocytes is based on co-assembly with *Xenopus*-endogenous CNNMs. Nevertheless, the effect is ARL15-specific, since co-expression of the related ARL8A protein, as suggested by the reviewers, did not alter TRPM7 currents. This was added to the revised manuscript (new Figure 3—figure supplement 2).

Reviewer #2:To understand the function and regulation of ion channels in the plasma membrane, knowledge of other proteins that are associated with channels in functional complexes is important. Here, Kollewe et al., used high resolution proteomics to assess the composition of native complexes containing TRPM7, a plasma membrane cation channel involved in cation transport, and containing a functional kinase in its C terminus.Using affinity purification and mass spectrometry, they convincingly identify a specific interaction of TRPM7 with members of the CNNM family of putative magnesium transporters and with the small G protein ARL15. This interaction can be reproduced in heterologous expression system. Moreover, in *Xenopus oocytes*, increasing levels of ARL15 lead to a dose-dependent inhibition of TRPM7 currents.Overall, this work represents an important next step in TRPM7 channel research, by systematically identifying the composition of the channel interactome, and how it affects channel functionality. The study can serve as a paradigm for similar studies on other (TRP) channels, which may yield a wealth of information on channel functional regulation.At this point, the (patho)physiological relevance of the interactions with ARL15 and CNNM1-4 remain unclear. Further work is needed to pinpoint potential effects of these proteins on TRPM7-dependent (magnesium) ion transport or kinase activity.

We appreciate the note of Reviewer 2 about the stimulatory impact of our study.

Overall, this is meticulous study identifying potentially important interaction partners of the cation channel TRPM7. A weaker point of the paper is that the functional effects of the interaction partners are studied/described in a somewhat superficial manner:1) The analysis of the currents measured in *Xenopus oocytes* only includes currents at +80 mV, which is far from the physiological range. It would be of interest to also provide additional data regarding the inward currents and, especially seen the interaction with CNNMs, the magnesium permeability of the complex.

Thank you for this important question. TRPM7 inward currents at negative membrane potentials are very small and prone to errors in quantitative assessment, for instance, due to the presence of endogenous Cl^-^ currents. Therefore, quantification of the comparably large TRPM7 outward currents is commonly used in the field. Nevertheless, to address the concern of Reviewer 2, we analysed TRPM7 inward currents in different experimental settings (new Figure 3—figure supplement 3, new Figure 4, new Figure 4—figure supplement 1). Of note, these additional analyses corroborate our conclusions derived from assessments of TRPM7 outward currents.

2) TRPM7 is a channel and an enzyme. However, the authors only investigate the effects of the binding partners on channel activity, not on kinase activity. Would it be possible to include an assay investigating the effect of ARL15 or CNMMs on kinase activity? In any case, such a potential regulatory effect should be discussed.

We agree with Reviewer 2 that the functional assessment of the TRPM7 kinase in the presence of CNNMs and ARL15 would benefit the impact of our study. It should be stressed that a kinase assay for the full-length TRPM7 protein was not available to address this question in the initial version of the paper. Therefore, we made substantial efforts to develop such an assay. We tested the *anti*-TRPM7 antibody, which specifically recognizes the known autophosphorylation site (Ser1511) in TRPM7 and TG100-115, a drug-like TRPM7 kinase inhibitor (new Figure 5—figure supplement 1). We observed that exposure of living cells to TG100-115 led to suppression of (p)S1511 TRPM7 immunoreactivity in a concentration- and time-dependent fashion. We concluded that detecting (p)Ser1511 TRPM7 levels is a reliable means to monitor TRPM7 kinase activity. Consequently, we investigated whether *ARL15* would modulate TRPM7 kinase activity and found no changes in (p)Ser1511 TRPM7 immunoreactivity (new Figure 5A). However, co-expression of CNNM3 resulted in a significant reduction of the (p)Ser1511 TRPM7 signal (new Figure 5), compatible with the notion that CNNM3 acts as a negative regulator of the TRPM7 kinase.

Labeling in Figure 1B seems incorrect.

The labelling of Figure 1B was corrected.

Reviewer #3:Chubanov et al., applied multi-epitope affinity purification and high-resolution quantitative MS analysis and identified TRPM7 forming a ternary complex with CNNM proteins and ARL15. CNNM1-4 proteins, like TRPM7, are thought to be magnesium transporters, so this finding is very intriguing as it relates to the mechanisms by which TRPM7 and CNNM, either directly or indirectly regulate magnesium homeostasis. The authors confirm these interactions with overexpression co-IP experiments in HEK cells and then evaluate the effects of ARL15 expression on TRPM7 function in *Xenopus oocytes*. They reveal that ARL15 is a negative regulator of TRPM7 channel function while maintaining TRPM7 plasma membrane localization. These studies are overall well performed, and technically rigorous. These initial findings are also very interesting and potentially impactful as they may finally explain a longstanding controversy about how TRPM7 (and TRPM6) may regulate magnesium homeostasis.

We thank Reviewer 3 for an overall positive evaluation of our study.

1. The finding that TRPM7 forms a complex with CNNM proteins, another putative magnesium transporter, now with a recent cryo-EM structure that supports a role in magnesium transport, is a very important finding that deserves to be further explored in this paper. As the authors recognize, the connection of TRPM7 to CNNM proteins has been previously described as has the connection between CNNM proteins and ARL15, which reduces novelty. Moreover, this manuscript appears to end at these initial findings which would normally be the beginning of a more impactful and meaningful paper.

We would like to emphasize that our manuscript does not present original data related to the interaction of CNNMs with ARL15 and cryo-EM structures of CNNMs, and we do not foresee how publications on this topic can compromise the novelty of our study. Moreover, after careful re-assessment of the current literature, we found no published works showing the direct interaction of CNNMs with TRPM7.

In the Discussion section, we refer to two manuscripts reporting proteome-wide screens performed in HEK293 and HTC116 cell lines using lentiviral overexpression of tagged proteins. These experiments proposed the interaction of recombinant TRPM7 with ARL15 but not with CNNMs (CNNMs were mentioned mistakenly in our text and we corrected it in the revised paper accordingly). By its experimental design, both screens only suggest a link between TRPM7 and ARL15 – which does, however, neither indicate direct interaction nor does it show co-assembly of native TRPM7 and ARL15 into macromolecular complexes in the rodent brain as identified de novo in this study. In addition, several other TRPM7 interactions were suggested in both screens but not confirmed by our experiments. Collectively, we do not think that the current scientific literature compromises the novelty of our findings.

However, we agree with Reviewer 3 that the assembly of CNNMs and ARL15 with TRPM7 is an intriguing finding that deserves further functional analysis. Accordingly, we conducted additional electrophysiological and biochemical experiments to elucidate the functional impact of ARL15 and CNNM3 on TRPM7. The obtained results further supported our initial hypothesis that ARL15 acts as a potent negative regulator of the TRPM7 channel. In contrast, our systematic assessment of CNNM3 effects on the channel and kinase moiety of TRPM7 revealed that CNNM3 represents the first known protein, which acts as a negative regulator of TRPM7 kinase activity. Both claims are novel and undoubtedly will stimulate follow-up studies across different research areas.

2. Relevant to the point above, how are CNNM protein expression and localization altered in TRPM7-/- cells? Could CNNM mistrafficking or impaired function account for the differences in magnesium homeostasis observed in TRPM7-/- cells?

As explained above (Essential revision point 4), we used ICP-MS to measure total concentrations of magnesium in TRPM7^-/-^ HEK293T cells transfected with TRPM7, CNNM3 or TRPM7 with CNNM3 cDNAs (Figure 4—figure supplement 2). We found that the ability of TRPM7 to regulate cellular Mg^2+^ levels was not affected by CNNM3.

It is important to note that the amount of CNNM proteins in the rodent brain exceeds that of TRPM7 by a factor of 400 as determined by quantitative MS-analysis. This means that only a minor portion of CNNMs occurs in complex with TRPM7 (while all TRPM7 proteins may co-assemble with CNNMs). We, therefore, did not investigate the localization of CNNM proteins in TRPM7^-/-^ HEK293Tcells.

It should also be stressed that the current literature provides conflicting views on the functional role of CNNMs. While some investigators suggested that mammalian CNNMs can function as Na^+^/Mg^2+^ exchanger responsible for Mg^2+^ efflux from the cell, other researchers proposed that CNNMs are responsible for Mg^2+^ uptake into cells. To the best of our knowledge, electrophysiological characterization of CNNM proteins was not achieved yet. Along these lines, resolving existing controversies over the functional role of CNNMs by using TRPM7 as a readout system may be misleading and is beyond the scope of the current study.

3. Ion channel functional studies are performed in *Xenopus oocytes* and the rationale is well explained with respect to titrating expression levels, but why not perform these studies in HEK cells to confirm that this holds true in the more relevant mammalian cells?

Thank you for this interesting question. As requested, we conducted patch-clamp experiments with HEK293 cells expressing CNNM3 or ARL15 (new Figure 3—figure supplement 4, new Figure 4—figure supplement 1). These experiments recapitulated our findings with *Xenopus oocytes*.

4. Are there differences in intracellular magnesium upon expression of TRPM7 +/- CNNM proteins +/- ARL15 in overexpression studies in mammalian cells?

As discussed above, we found no effect of CNNM3 on TRPM7 Mg^2+^ currents (new Figure 4—figure supplement 1) or total Mg levels in HEK293 cells (new Figure 4—figure supplement 2).

5. TRPM7 antibodies are notoriously challenging with respect to specificity. The authors described negative control experiments using TRPM7-/- cells. Where is the TRPM7 KO negative control data?

The negative controls presented as peak volumes, abundance_norm_ values, and ratio distance values have been included in the original manuscript as Supplementary file 1 to Figure 1.

6. The co-IP experiments with overexpression of TRPM7 and then each of identified TRPM7 interactors, CNNM1-4, ARL15 is the least specific experiment possible for confirmation. Why not perform endogenous TRPM7 IP (since the Abs seem to work for IP) and then probe for endogenous, or even overexpressed CNNM and ARL15? Conversely, is it possible to IP endogenous, or overexpressed CNNM or ARL15 and detect endogenous TRPM7 on WB? These additional co-IP would strengthen this finding.

As pointed out in (1) of *‘Essential revision’*, TRPM7 was first affinity-isolated from native tissue (rodent brain), and the identified interactors CNNMs and ARL15 were verified in subsequent experiments. Given the large excess of CNNM proteins over TRPM7 in membranes of cells/tissues tested so far, reverse purification does not appear to be a promising approach. Furthermore, it would require exceptionally efficient *anti-CNNM* antibodies and KO controls which are unfortunately not yet at hand.

7. The new phosphosites identified are interesting, but what is their functional significance?

We introduced phosphomimetic mutations in a subset of identified phospho-sites (S1208D, S1360D, S1480D, S1496D and S1567D) and found that three TRPM7 mutants (S1208D, S1496D and S1567D) displayed enhanced current amplitudes, suggesting that phosphorylation of TRPM7 may represent a new regulatory mechanism as recently shown for a related ion channel, i.e. TRPM8 (*PMID: 34446569* ). These results are shown in new figure 6.

8. For the experiments in Figure 4, ARL15 expression is titrated based on injected mRNA, but it would also be advisable to show the actual levels of overexpressed protein by WB.

We added such data in Figure 3C.